# Can MLLMs Absorb Math Reasoning Abilities from LLMs as Free Lunch?

**Yijie Hu**[1,2]*, **Zihao Zhou**[1,2]*, **Kaizhu Huang**[3], **Xiaowei Huang**[2], **Qiufeng Wang**[1]†

[1] Xi'an-Jiaotong Liverpool University
[2] University of Liverpool
[3] Duke Kunshan University

## Abstract

Math reasoning has been one crucial ability of large language models (LLMs), where significant advancements have been achieved in recent years. However, most efforts focus on LLMs by curating high-quality annotation data and intricate training (or inference) paradigms, while the math reasoning performance of multimodal LLMs (MLLMs) remains lagging behind. Since the MLLM typically consists of an LLM and a vision block, we wonder: *Can MLLMs directly absorb math reasoning abilities from off-the-shelf math LLMs without tuning?* Recent model-merging approaches may offer insights into this question. However, they overlook the alignment between the MLLM and LLM, where we find that there is a large gap between their parameter spaces, resulting in lower performance. Our empirical evidence reveals two key factors behind this issue: the identification of crucial reasoning-associated layers in the model and the mitigation of the gaps in parameter space. Based on the empirical insights, we propose **IP-Merging** that first **I**dentifies the reasoning-associated parameters in both MLLM and Math LLM, then **P**rojects them into the subspace of MLLM, aiming to maintain the alignment, and finally merges parameters in this subspace. IP-Merging is a tuning-free approach since parameters are directly adjusted. Extensive experiments demonstrate that our IP-Merging method can enhance the math reasoning ability of MLLMs directly from Math LLMs without compromising their other capabilities. [3].

## 1 Introduction

As one fundamental ability of large language models (LLMs), improving math reasoning abilities is crucial to handle complex problem-solving tasks [1]. By creating a substantial amount of high-quality math reasoning data [47] and designing intricate training procedures, LLMs such as GPT-4 or Qwen have achieved remarkable progress in solving text-based math reasoning problems [1, 2, 9]. Despite these advancements, the challenge of mathematical reasoning remains a significant obstacle for MLLMs [19, 23, 57, 54]. Visual math reasoning tasks, which are essential for real-world applications, require MLLMs to extract image information, analyze problem constraints, integrate text and visuals, and perform complex reasoning.

Following similar training strategies for improving math reasoning abilities in LLMs, attempts have been made to enhance MLLM's math reasoning skills [30]. Despite the notable progress, collecting and annotating high-quality multimodal reasoning data [8, 56] is expensive. Additionally, training large models demands extensive computational resources, which makes it costly to improve their reasoning abilities. As MLLMs typically include a foundation LLM as the core component [20]

---

*Equal contribution.
†Corresponding author.
[3]Code Repository: `https://github.com/tambourine666/MergeVLM`

39th Conference on Neural Information Processing Systems (NeurIPS 2025).

and share similar training processes with LLMs, we propose one intriguing question: *Can MLLM directly absorb math reasoning abilities from off-the-shelf math LLMs to enhance the multi-modal math reasoning without tuning?*

To answer this question, our initial attempt is to adopt model merging techniques, which intend to integrate task-specific knowledge into one model by merging multiple fine-tuned models without involving any training [15, 41]. We illustrate this process in fig. 1, where we aim to improve MLLM's multi-modal math reasoning by integrating parameters from math LLM into MLLM. One representative method called task arithmetic [15] extracts the task vector of each model via the subtraction of the fine-tuned and pre-trained models (i.e., $\Delta \boldsymbol{W} = \boldsymbol{W}_{ft} - \boldsymbol{W}_{pre}$). By adding all task vectors, the merged model is expected to perform well on all tasks. However, the direct merging of the reasoning-associated LLM task vector and MLLMs does not improve the reasoning abilities of MLLMs (fig. 2(a)). We observe that existing merging methods work effectively when task vectors are similar [32, 45]. However, this does not hold between MLLMs and Math LLMs. We argue that there exists a gap between their task vectors. Without aligning them properly, direct addition leads to conflicts. MLLMs are designed to integrate visual and textual inputs, whereas math LLMs acquire reasoning skills from text-based mathematical problems. This fundamental difference creates a substantial discrepancy in their task vectors. To illustrate this, inspired by previous works [16, 39] that adopt trace value to quantify gradient change and learning difficulty in the fine-tuning stage, we plot the trace value of task

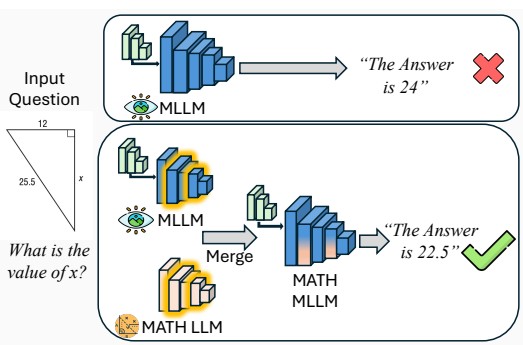

Figure 1: We aim to enhance multi-modal reasoning skills of MLLM by merging parameters in MLLM and Math LLM without tuning or changing the size of the model.

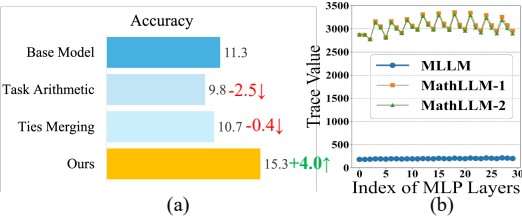

(a)  (b)

Figure 2: (a) Comparison of different methods of merging MLLM with math LLM on MathVerse. (b) Trace value of task vectors in 30 MLP layers in two math LLMs (MetaMath-Llemma-7B and Tora-code-7B) and one MLLM (LLaVA-v1.5-7B).

vectors in 30 MLP blocks (fig. 2(b)), and there is a large gap between the MLLM and Math LLM.

To delve into the alignment issue in integrating reasoning abilities from math LLMs into MLLMs, we conduct empirical analysis and unveil two key challenges: (1) *How to identify math reasoning abilities associated parameters in MLLM and LLM?* We demonstrate that math reasoning-associated parameters appear highly similar in the subspace; absorbing these parameters improves multi-modal math reasoning, detailed in section 4.1. (2) *How to quantify and mitigate the gaps between models in the parameter space?* We illustrate that parameter gaps between models can be quantified by singular values, detailed in section 4.2. Bridging these gaps can enhance the alignment between models and improve math reasoning performance.

Based on our empirical findings, we propose **IP-merging** that **I**dentifies the math reasoning parameters in the models, then **P**rojects math reasoning parameters into the subspace of MLLM for better alignment. In the parameter selection stage, the selection process is guided by the similarity between the subspaces of the task vectors from the two models. To achieve this, we first perform singular value decomposition (SVD) on the task vectors of both the LLM and the MLLM. Next, we compute the cosine similarity between the basis vectors extracted from the LLM and the MLLM. Parameters corresponding to the most similar subspaces are then selected for the subsequent merging process. To further align the selected parameters, these parameters are rescaled by computing the rescaling factor based on the eigenvalues of the selected parameters in MLLM and math LLM. The rescaled math LLM parameters are projected into the subspace of the MLLM, allowing the projected LLM parameters to be close to the MLLM parameters. Finally, the rescaled parameters are merged into the MLLM. The overall process allows the reasoning capabilities of the LLM to be effectively transferred

and adapted to the multimodal context of the MLLM. Our method requires no data or any additional tuning and can be efficiently implemented.

We demonstrate the effectiveness of our method by merging MLLM (e.g., LLaVA series and Qwen series) and different math reasoning LLMs. We validate the performance of our method on MathVista, MathVerse, DynaMath and MathVision for evaluating math reasoning abilities. We further show our method does not interfere with the model's other abilities by evaluating our method general knowledge datasets, i.e., MMMU [50], TextVQA [31] and MMBench [22]. Our contributions can be summarized as follows:

- We propose the problem of improving the math reasoning abilities of MLLMs by directly absorbing math LLMs without any tuning.

- We reveal two key challenges to answer this question, including the selection of math-reasoning-associated parameters and the reduction of gaps between models.

- We propose the IP-merging method, which first identifies the crucial layers and then aligns the selected layers by projecting them into the MLLMs subspace.

- Extensive experimental results validate that the IP-Merging method enhances the math reasoning abilities of MLLMs without compromising other capabilities.

## 2 Related Works

### 2.1 Math Reasoning of MLLMs

The community has made significant efforts to improve mathematical reasoning ability, which is regarded as a fundamental capability of MLLMs. In the pre-training stage, previous works focus on enhancing math reasoning of base models by collecting math pre-training data [37, 11], generating synthetic data [5], and optimizing training strategies [18]. Furthermore, in the post-training phase, researchers significantly increase the scale and quality of math reasoning instruction data through data augmentation techniques [8, 12, 30]. The introduction of variant fine-tuning [55], reinforcement learning algorithms [48, 49, 38], and self-evolving frameworks [51, 21] greatly improves the efficiency of data utilization. Recently, O1-like models successfully leverage the deep-thinking chain of thought for inference scaling [6], substantially advancing reasoning capabilities. Different from these cost-intensive works (either training or inference), we aim to improve MLLMs' math reasoning abilities by directly absorbing them from Math LLMs without any tuning.

### 2.2 Model Merging

Model merging has emerged as a promising technique for enhancing the capabilities of models without requiring raw training data or intensive computation, which offers a low-cost solution to elevate the abilities of LLMs. Model merging takes off-the-shelf task-specific models and fuses all models into a single model with diverse abilities [43]. Several advanced methods have been developed for model merging, which can be broadly classified into pre-merging and during-merging strategies. Pre-merging methods focus on merging model weights [40, 15, 41], architecture transformations [32, 52], or disentangling weight spaces to create optimal conditions for merging [33, 25]. During-merging methods address task conflicts using basic averaging, weighted strategies, subspace projections, dynamic routing, or post-merging calibrations [46, 45, 44]. Previous methods focus on merging models trained on single modality [43, 17], while our method aims to merge models from different modalities.

## 3 Model Merging Framework

### 3.1 Preliminaries of Task Vectors

Let $\boldsymbol{W}_0$ denote the parameters of a pre-trained language model, such as Llama-based model [34]. The math reasoning model $\boldsymbol{W}_{Math}$ is obtained by fine-tuning the pretrained model $\boldsymbol{W}_0$ on math reasoning data $\mathcal{D}_{math-txt}^{train}$. Math task vectors are defined as the difference between parameters of LMs before and after finetuning, i.e., $\Delta \boldsymbol{W}_{Math} = \boldsymbol{W}_{Math} - \boldsymbol{W}_0$. Here, the task vectors are

obtained by subtraction at each layer. For example, if the model has $N$ layers, the task vector is $\Delta \boldsymbol{W}_{Math} = \{\Delta \boldsymbol{W}^1_{Math}, \Delta \boldsymbol{W}^2_{Math}, \ldots, \Delta \boldsymbol{W}^N_{Math}\}$.

For an MLLM such as LLaVA [20], the model is trained by freezing the visual encoder [28] and tuning the projection layers and the LLM using vision and language pairs $\mathcal{D}^{train}_{vl}$. Here, we denote the models' task vector of receiving multi-modal input as $\Delta \boldsymbol{W}_{MLLM} = \boldsymbol{W}_{MLLM} - \boldsymbol{W}_0$. The task vector of MLLM is computed between the LLM and the pretrained LLM, which does not include the frozen visual encoder and the visual projection layers. During the model merging process, the visual encoder and visual projection layers are not modified, only LLMs in the models are merged.

## 3.2 Model Merging Framework

Model merging techniques fuse several fine-tuned task-specific models into one comprehensive multi-task model without training the models. In our case, we aim to obtain one math reasoning MLLM $\boldsymbol{W}_{MathMLLM}$ by merging math LLM $\boldsymbol{W}_{Math}$ with the MLLM $\boldsymbol{W}_{MLLM}$. For simplicity, we consider the case of merging two models here. We can formulate the general framework for merging the models as

$$\boldsymbol{W}_{MathMLLM} = \mathcal{F}(\boldsymbol{W}_0, \Delta \boldsymbol{W}_{MLLM}, \Delta \boldsymbol{W}_{Math}; \mathbf{M}). \tag{1}$$

Prevailing approaches, such as Task Arithmetic [15], Ties Merging [41], EMR Merging [14] can be further formulated as

$$\boldsymbol{W}_{MathMLLM} = \boldsymbol{W}_0 + \alpha_1 f_1(\Delta \boldsymbol{W}_{MLLM}, \mathbf{M}_1) + \alpha_2 f_2(\Delta \boldsymbol{W}_{Math}, \mathbf{M}_2), \tag{2}$$

where $\alpha_1$ and $\alpha_2$ are scaling parameters, $\mathbf{M}_1$ and $\mathbf{M}_2$ can be regard as alignment matrices. $f_1(\cdot)$ and $f_2(\cdot)$ represent two mapping functions, such as the dot product of the element-wide product. For example, task arithmetic [15] set $\mathbf{M}_1$ and $\mathbf{M}_2$ to identity mapping. Ties merging [41] and DARE [46] compute sparse matrices $\mathbf{M}_1$ and $\mathbf{M}_2$, and select parameters by element-wise product. EMR merging computes scaling factors $\alpha_1$, $\alpha_2$ based on the absolute values and derives task-specific masks $\mathbf{M}_1$, $\mathbf{M}_2$. Our method follows a similar model merging framework. After merging the models, the model is then tested on multi-modal math reason dataset $\mathcal{D}^{test}_{math-vl}$ to evaluate its performance.

# 4 Methodology

In this section, we delve into the alignment issue in model merging by identifying parameters associated with math reasoning in section 4.1, and then we quantify the gaps between models in section 4.2. Finally, we describe our proposed IP-merging in section 4.3.

## 4.1 Identify Math-Reasoning-Associated Parameters

Experimental results in fig. 2 provide one critical observation: though limited, the MLLM already demonstrates an inherent capability for multi-modal mathematical reasoning. As the MLLM is trained to align visual and textual input, the direct addition of task vectors in all layers derived from a math reasoning LLM harms the learned alignment, yielding suboptimal performance. To address this issue, a key question arises: *Which parameters are associated with mathematical reasoning abilities in MLLM, and how can we effectively identify and prioritize them for merging?*

To answer this question, we propose to quantify this correlation by calculating the subspace similarity between the parameters in Math LLMs and MLLM. Recent efforts [26, 24] demonstrate that task-specific competencies, such as mathe-

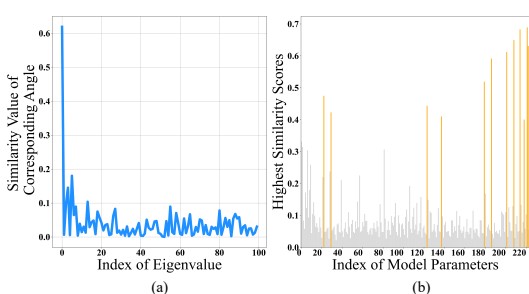

Figure 3: (a) Absolute cosine value of the top 100 corresponding angles in one MLP layer. (b) The highest similarity score distribution among model parameters, orange bars represent the parameters with a high similarity score.

matical reasoning abilities, reside within particular subspaces of the model's parameter space. If the

subspace of one parameter in MLLM has higher subspace similarity with math LLM, this parameter can be regarded as crucial for math reasoning abilities. Specifically, given the math reasoning task vector $\Delta \boldsymbol{W}_{Math}$ and multi-modality task vector $\Delta \boldsymbol{W}_{MLLM}$ in the layer $n$, we can first factorize these weights using singular value decomposition (SVD) respectively:

$$\Delta \boldsymbol{W}_{Math}^n = \mathbf{U}_M^n \boldsymbol{\Sigma}_M^n \mathbf{V}_M^{n\top}, \quad \Delta \boldsymbol{W}_{MLLM}^n = \mathbf{U}_V^n \boldsymbol{\Sigma}_V^n \mathbf{V}_V^{n\top}, \tag{3}$$

which can be organized into a combination of singular values and orthonormal bases:

$$\Delta \boldsymbol{W}_{Math}^n = \sum_{i=1}^d \sigma_{M,i}^n \boldsymbol{u}_{M,i}^n \boldsymbol{v}_{M,i}^{n\top}, \quad \Delta \boldsymbol{W}_{MLLM}^n = \sum_{i=1}^d \sigma_{V,i}^n \boldsymbol{u}_{V,i}^n \boldsymbol{v}_{V,i}^{n\top}, \tag{4}$$

where $\sigma_{V,i}^n$ and $\sigma_{M,i}^n$ represent the $i$-th singular value of $n$-th layer in MLLM and Math-LLM respectively, and $d$ is the number of dimensions. $\sigma_{M,1}^n \geq \sigma_{M,2}^n ... \geq \sigma_{M,d}^n$, vice versa for $\sigma_{V,i^n}$. $\boldsymbol{v}_{M,i}^n, \boldsymbol{v}_{M,i}^n, \boldsymbol{u}_{V,i}^n \boldsymbol{v}_{V,i}^n$ represent corresponding orthonormal bases respectively. If the direction of two sets of orthonormal bases is closely aligned, the similarity between the two subspaces is high. Meanwhile, we consider the singular value as the reference when computing similarity, as the singular values determine the importance of each orthonormal basis. To this end, we propose to use the corresponding angle to measure the similarity between two subspaces [4]:

**Definition 1** (Similarity Value of Corresponding Angle). *Given two groups of eigenvectors:* $\{\boldsymbol{v}_{M,1}^\top, \ldots, \boldsymbol{v}_{M,d}^\top\}$ *and* $\{\boldsymbol{v}_{V,1}^\top, \ldots, \boldsymbol{v}_{V,d}^\top\}$, *the corresponding angle represents the angle between two eigenvectors corresponding to the same eigenvalue index. The cosine value of the $i$-th eigenvector's corresponding angle is*

$$S_i^n = \frac{\langle \boldsymbol{v}_{M,i}^{n\top}, \boldsymbol{v}_{V,i}^{n\top} \rangle}{\left\| \boldsymbol{v}_{M,i}^{n\top} \right\| \cdot \left\| \boldsymbol{v}_{V,i}^{n\top} \right\|}. \tag{5}$$

We visualize the distribution of the first 100 corresponding angles in a selected MLP layer in fig. 3(a). We can see that the cosine value of the first corresponding angle is significantly higher than those of the other angles. This observation suggests that the basis associated with the largest singular value represents a subspace in MLLM that is strongly linked to math reasoning capabilities. Consequently, the parameters with high subspace similarity may play a more critical role in math reasoning abilities.

To explore this further, we plot the distribution of the highest similarity value scores across model layers in fig. 3(b). The figure illustrates that parameters have high similarity scores, which can be regarded as crucial to math reasoning. To validate their importance, we perform experiments where we selectively merge the corresponding layers of the LLM into these parameters of the MLLM with a high similarity score in table 1. The gain of performance verifies the critical role of the identified MLLM parameters in facilitating math reasoning. This finding shows that subspace similarity can be one key criterion for selecting and prioritizing math reasoning parameters. More analysis can be found in appendix D.

Table 1: Performance comparison after selecting reasoning-related layers to merge on MathVerse.

| Approach | Average Accuracy |
|---|---|
| Base MLLM | 11.3 |
| Task Arithmetic | 9.8 $^{-1.5\downarrow}$ |
| Task Arithmetic+Param Selection | **13.4** $^{+2.1\uparrow}$ |

## 4.2 Gaps between Models in Parameter Space

As we aim to absorb math reasoning abilities from LLMs to MLLMs, the gap between models emerges as a significant obstacle. While both models are derived from the same foundational LLM (e.g., LLaMA), the parameter changes within MLLMs and math reasoning LLMs can vary significantly. This disparity reflects fundamental differences in their learned representations, task objectives, and domain-specific knowledge. To quantify these differences, we use the distribution of eigenvalues in the model parameters as an intuitive metric for understanding the magnitude and nature of these changes. Specifically, the eigenvalue distribution offers insight into the "scale" of parameter updates. We plot the top 1024 singular value distributions of one MLP layer in the model fig. 4(a), which reveals that LLMs and MLLMs exhibit different eigenvalues, highlighting the inherent gap between LLMs and MLLMs. When combining models of a math reasoning LLM with large eigenvalues and an MLLM with smaller eigenvalues by addition, parameters with larger eigenvalues can overshadow the contributions of parameters with smaller ones. Here, we take task arithmetic as an example:

**Proposition 1.** *Let $\Delta \boldsymbol{W}_{Math}$ and $\Delta \boldsymbol{W}_{MLLM}$, be two real matrices. The L2-norm of two matrices is defined by $\|\Delta \boldsymbol{W}_{Math}\|_2 = \sigma_{\max}(\Delta \boldsymbol{W}_{Math}), \|\Delta \boldsymbol{W}_{MLLM}\|_2 = \sigma_{\max}(\Delta \boldsymbol{W}_{MLLM})$, Model merged by task arithmetic satisfies the triangle's inequality [13]*

$$\|\Delta \boldsymbol{W}_{Math}\|_2 - \|\Delta \boldsymbol{W}_{MLLM}\|_2 \leq \|\Delta \boldsymbol{W}_{Math} + \Delta \boldsymbol{W}_{MLLM}\|_2 \leq \|\Delta \boldsymbol{W}_{Math}\|_2 + \|\Delta \boldsymbol{W}_{MLLM}\|_2,$$

*If we assume*

$$\|\Delta \boldsymbol{W}_{MLLM}\|_2 \leq \varepsilon \|\Delta \boldsymbol{W}_{Math}\|_2, \quad 0 < \varepsilon < 1,$$

*then,*

$$(1 - \varepsilon) \|\Delta \boldsymbol{W}_{Math}\|_2 \leq \|\Delta \boldsymbol{W}_{Math} + \Delta \boldsymbol{W}_{MLLM}\|_2 \leq (1 + \varepsilon) \|\Delta \boldsymbol{W}_{Math}\|_2.$$

Our proposition demonstrates that if the maximum singular value of the math reasoning parameters in LLM is far larger than the MLLM parameter, i.e., $\sigma_{\max}(\Delta \boldsymbol{W}_{Math}) \gg \sigma_{\max}(\Delta \boldsymbol{W}_{MLLM})$, $\varepsilon \to 0$, the MLLM parameter will be overshadowed by math parameters ($\|\Delta \boldsymbol{W}_{Math} + \Delta \boldsymbol{W}_{MLLM}\|_2 \approx \sigma_{\max}(\Delta \boldsymbol{W}_{Math})$). To validate this point, we further plot the distribution of the maximum singular across model parameters in fig. 4(b), indicating that the Math LLM model may overshadow MLLM during merging. Therefore, it is crucial to align the LLM and MLLM in the parameter space to allow the smooth transfer of math reasoning abilities.

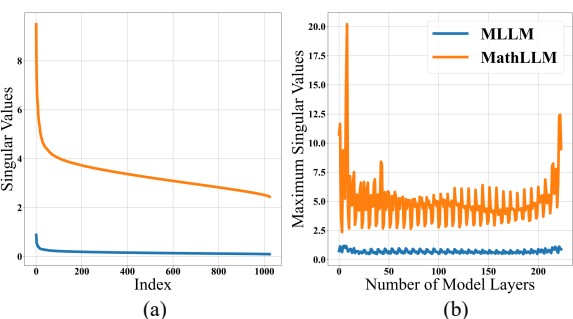

Figure 4: (a) Singular values distribution in one MLP layer. We plot the top 1024 singular values. (b) Maximum singular values across all model parameters.

### 4.3 IP-Merging

To transfer reasoning abilities from LLMs to MLLMs, we propose a novel merging method called IP-merging. Our method first identifies the crucial parameters in both MLLM and math LLM, then projects the rescaled selected LLM parameters into the MLLM subspace for better alignment. Finally, the aligned parameters are merged with MLLM. In the parameter identification stage, crucial math-reasoning-associated parameters between LLMs and MLLMs are identified and selected based on their similarity in the subspace. In the parameter projection stage, these selected parameters are rescaled, aligned, and projected into the MLLM subspace to minimize the gap between the models. The overall process can be found in algorithm 1 and appendix F.

#### 4.3.1 Parameter Identification

Given that MLLMs share a larger number of layers with LLMs, we restrict the merging process to components shared by both models, such as attention layers and MLP layers, without modifying the visual encoder or projection layer. Let $N$ denote the total number of layers considered for merging. To identify compatible parameters, we compute the corresponding angle for each layer $\{S_1^n, S_2^n, \ldots, S_d^n\}$ between the MLLM and Math LLM using a cosine similarity metric (refer to definition 1). Parameters with a similarity score higher than the threshold $S_\alpha$ are selected for merging, while others are excluded. This selection process can be formalized as follows

$$\Delta \bar{\boldsymbol{W}}_{Math}^n = \begin{cases} \Delta \boldsymbol{W}_{Math}^n & S_1^n \geq S_\alpha \\ 0 & \text{otherwise} \end{cases}. \tag{6}$$

#### 4.3.2 Parameter Projection

Empirical observations in section 4.2 highlight the importance of aligning parameters across different model modalities. To facilitate the alignment, Ties Merging [41] or EMR Merging [14] offers a straightforward solution by selecting parameters with consistent signs (i.e., both positive or both negative) and discarding the rest. While these works reduce interference between the merged models, they risk discarding critical parameters, leading to a decline in overall performance as shown in fig. 2.

Different from previous works, we aim to merge the parameters of Math LLM into the MLLM without harming other abilities of MLLM. To achieve this, our key idea is to reduce the gap between LLM and MLLM parameter spaces by rescaling and projecting the selected mathematical reasoning parameters from the LLM into the subspace of the MLLM. By rescaling the layers of LLM, we can reduce parameter gaps between the models. We further project the selected LLM layers into the subspace of MLLM, which pushes parameters in LLM to lie close to the MLLM in the parameter space, fostering the alignment between the two models [29].

**Rescaling.** To normalize the parameter magnitudes between the two models, we compute a rescaling factor $\lambda_n$, defined as the ratio of the nuclear norms of the corresponding parameter spaces in the MLLM and LLM

$$\lambda_n = \frac{\sum_{i=1}^{d} \sigma_{V,i}^n}{\sum_{i=1}^{d} \sigma_{M,i}^n}, \tag{7}$$

where $\sigma_{V,i}^n$ and $\sigma_{M,i}^n$ are the singular values of the corresponding parameters in the MLLM and LLM, respectively.

**Projection.** To align the rescaled parameters to the MLLM subspace, we calculate an importance score $\gamma_n$ for the subspace vectors based on their similarity $S_i^n$. The importance score is defined as

$$\gamma_i^n = \frac{exp(S_i^n)}{\sum_{i=1}^{d} exp(S_i^n)}. \tag{8}$$

Using these importance scores, the selected mathematical reasoning layers from the LLM are projected into the MLLM weighted subspace as

$$\bar{\boldsymbol{V}}_V^n = \gamma_n \mathbf{V}_V^{n\top}, \quad \Delta \boldsymbol{W}_{Math-P}^n = \lambda_n \Delta \bar{\boldsymbol{W}}_{Math}^n \bar{\boldsymbol{V}}_V^n \bar{\boldsymbol{V}}_V^{n\top}. \tag{9}$$

This projection emphasizes the subspace basis vectors that are most relevant to mathematical reasoning, thereby ensuring effective alignment of the LLM parameters with the MLLM architecture. After projection, we can obtain the math reasoning MLLM by adding the projected parameter to MLLM as

$$\boldsymbol{W}_{MathMLLM}^n = \boldsymbol{W}_0^n + \Delta \boldsymbol{W}_{MLLM}^n + \lambda_n \Delta \bar{\boldsymbol{W}}_{Math}^n \bar{\boldsymbol{V}}_V^n \bar{\boldsymbol{V}}_V^{n\top}, \tag{10}$$

By referring to eq. (2), our method also fits the general framework, where $\alpha_1$ is sets as 1 and $\alpha_2$ is set as $\lambda_n$, $f_1(\cdot, \mathbf{M}_1)$ is the identity mapping, and $f_2(\cdot, \mathbf{M}_2)$ is the projection matrix in eq. (9), the overall algorithm of the proposed method can be found in appendix F.

## 5 Experiments

### 5.1 Experiments Setup

We test our models on six benchmarks, i.e., MathVista [23], MathVerse [53], DynaMath (DM) [59], MathVision [36] and three general QA benchmarks MMMU [50], TextVQA [31] and MMBench [22]. MathVista can be divided into five subsets: Figure Question Answering (FQA), Geometry Problem Solving (GPS), Math Word Problems (MWP), Textbook Question Answering (TQA), and Visual Question Answering (VQA). MathVerse includes a diverse set of math problems that require reasoning over both textual and visual information, such as charts, diagrams, and equations, which can be divided into five subsets, i.e., Text Dominant (T-D), Text Lite (T-L), Vision Intensive (V-I), Vision Dominant (V-D) and Vision Only (V-O). DynaMath (DM) is a dynamic visual math benchmark designed for in-depth assessment of VLMs. MathVision (Math-V) is a collection of high-quality mathematical problems with visual contexts sourced from real math competitions. The MMMU benchmark is suitable for assessing the general knowledge of MLLM. TextVQA evaluates a model's general ability to read and reason about text in images, requiring joint understanding of visual content and language reasoning. MMBench is a comprehensive multimodal benchmark that tests large models across diverse tasks to assess their general multimodal intelligence and robustness. We select the English split for evaluation. We compare our proposed methods with prevailing model merging techniques such as Task Arithmetic [15], Ties Merging [41] and EMR Merging [14]. More details on implementation and hyperparameters can be found in appendix B and appendix B.3, respectively.

Table 2: Comparison with other model merging approaches on the MathVerse and MathVista. **Bold** represents the best performance. We merge LLaVA with Tora-code-7B, Qwen2-VL-7B-Instruct with Qwen-2-Math-7B models, InternVL3-8B-Instruct with DeepSeek-R1-distilled-Qwen-7B, respectively.

| Approach | MathVerse | | | | | | MathVista | | | | | | DM | Math-V |
|---|---|---|---|---|---|---|---|---|---|---|---|---|---|---|
| | T-D | T-L | V-I | V-D | V-O | Overall | TQA | GPS | VQA | FQA | MWP | Overall | | |
| *LLaVA-1.5-7B as Base Model* | | | | | | | | | | | | | | |
| Base Model | 12.4 | 10.0 | 12.2 | 12.8 | 9.1 | 11.3 | 36.1 | 22.1 | 37.4 | 20.8 | 13.9 | 25.2 | 15.4 | 11.1 |
| Task Arithmetic | 12.4 | 11.0 | 12.6 | 11.9 | 0.9 | 9.8 | 31.7 | 33.6 | 29.6 | 21.6 | 11.3 | 25.2 | 13.9 | 11.0 |
| Ties Merging | 13.1 | 14.0 | 13.3 | 13.1 | 0.1 | 10.7 | 39.9 | **40.4** | 26.8 | 23.8 | 6.5 | 27.1 | 14.0 | 10.6 |
| EMR Merging | 10.8 | 10.7 | 11.3 | 12.6 | 6.7 | 10.4 | 36.7 | 25.9 | 30.7 | 21.6 | 13.9 | 25.0 | 13.8 | 11.6 |
| IP-Merging | **16.0** | **16.1** | **14.1** | **15.5** | **15.0** | **15.3**$^{4.0↑}$ | **43.7** | 21.6 | **40.8** | **24.9** | **15.1** | **28.2**$^{3.0↑}$ | **16.1**$^{0.7↑}$ | **11.8**$^{0.7↑}$ |
| *Qwen-2-VL-7B-Instruct as Base Model* | | | | | | | | | | | | | | |
| Base Model | 27.4 | 26.8 | 27.3 | 25.6 | 16.9 | 24.8 | 58.9 | 33.7 | **58.7** | 66.5 | 57.5 | 55.4 | 40.8 | 16.3 |
| Task Arithmetic | 15.7 | 10.9 | 10.0 | 9.8 | 0.3 | 9.3 | 37.3 | 38.0 | 32.4 | 25.3 | 25.3 | 31.1 | 14.8 | 11.3 |
| Ties Merging | 6.9 | 6.5 | 7.0 | 6.2 | 0.1 | 5.3 | 39.9 | 38.5 | 32.4 | 21.6 | 18.3 | 29.3 | 15.1 | 11.4 |
| EMR Merging | 20.9 | 18.4 | 18.3 | 18.4 | 11.9 | 17.6 | 50.6 | 35.6 | 44.7 | 43.1 | 30.1 | 40.8 | 21.8 | 14.5 |
| IP-Merging | **31.0** | **28.7** | **29.4** | **29.7** | **23.6** | **28.5**$^{3.7↑}$ | **63.3** | **41.8** | 57.5 | **69.9** | **66.7** | **60.2**$^{4.8↑}$ | **41.0**$^{0.2↑}$ | **19.1**$^{2.8↑}$ |
| *InternVL3-8B-Instruct as Base Model* | | | | | | | | | | | | | | |
| Base Model | 47.6 | 40.9 | 39.1 | **37.8** | 27.5 | 38.5 | **65.2** | 70.2 | **53.1** | 65.8 | 75.3 | 66.1 | 50.7 | 24.9 |
| Task Arithmetic | 24.9 | 18.5 | 15.6 | 14.1 | 11.2 | 16.9 | 43.0 | 46.2 | 26.8 | 20.1 | 29.0 | 32.0 | 20.1 | 15.3 |
| Ties Merging | 18.8 | 13.8 | 14.6 | 15.0 | 14.2 | 15.3 | 34.8 | 34.6 | 25.1 | 19.7 | 13.4 | 25.0 | 16.1 | 9.6 |
| EMR Merging | 35.9 | 30.7 | 29.4 | 27.3 | 27.3 | 30.1 | 59.5 | 51.4 | 42.5 | 45.0 | 50.5 | 49.2 | 39.2 | 18.8 |
| IP-Merging | **48.1** | **41.4** | **39.5** | 37.6 | **28.3** | **39.0**$^{0.5↑}$ | 63.9 | **74.5** | 52.5 | **68.2** | **76.8** | **67.6**$^{1.6↑}$ | **51.4**$^{0.7↑}$ | **25.2**$^{0.3↑}$ |

## 5.2 Experimental Results

We conduct our experiments on MathVista and MathVerse in table 2, and MMMU in table 3. For LLaVA models, we select LLaVA-1.5-7B as the base MLLM and Tora-Code-7B [9] as the math reasoning LLM, as it is trained on logical reasoning-based math problems. For Qwen models, we select Qwen-2VL-7B-Intruct as the base MLLM and Qwen-2-math-7B [42] as the math LLM. For InternVL models, we select InternVL3-8B-Instruct [58] as the base MLLM and DeepSeek-R1-distilled-Qwen-7B as the math LLM.

Compared to existing merging methods, our approach consistently improves performance across all sub-tasks. Notably, it achieves at least a 3.7% average gain over the original model on MathVerse, demonstrating its effectiveness in enhancing general math reasoning capabilities. For MathVista, the biggest improvement for LLaVA models lies in the task of the TQA subset, where science and math knowledge are highly demanded, improving the base model by 7.6% . The LLaVA model does not achieve a performance gain in GPS after merging, where the multiple steps of geometry reasoning are required. On the other hand, the ties merging boost the LLaVA performance in GPS, owing to its ability to resolve model conflicts by selecting parameters with the same sign. The consistent sign may indicate shared abilities across models. In this case, the visual reasoning ability is retained and enhanced by merging with the text-based reasoning ability in the Math LLM, but other general abilities deteriorate (a drop of 6.9% in MMMU) or VQA tasks (a drop of 6.7%). The Qwen model obtains the performance gain of 4.8% on average, with the most notable performance gain in MWP, where task-specific reasoning skills are highly demanded, improving the model by 9.2%.

Table 3: Comparison with other merging approaches on three general benchmarks. IP-merging preserves general abilities.

| Approach | MMMU | TextVQA | MMBench |
|---|---|---|---|
| *LLaVA-1.5-7B as Base Model* | | | |
| Base Model | 34.2 | 47.5 | 62.7 |
| Task Arithmetic | 30.2 | 23.5 | 17.4 |
| Ties Merging | 27.3 | 1.1 | 8.2 |
| EMR Merging | **34.8** | **48.0** | 57.1 |
| IP-Merging | 34.4 | 47.6 | **63.1** |
| *Qwen-2-VL-7B-Instruct as Base Model* | | | |
| Base Model | 50.7 | 83.8 | 80.3 |
| Task Arithmetic | 32.6 | 37.9 | 54.1 |
| Ties Merging | 30.8 | 4.3 | 43.7 |
| EMR Merging | 41.8 | 73.2 | 74.8 |
| IP-Merging | **50.7** | **84.0** | **80.8** |
| *InternVL3-8B-Instruct as Base Model* | | | |
| Base Model | 61.6 | 81.9 | 84.4 |
| Task Arithmetic | 32.6 | 18.9 | 31.3 |
| Ties Merging | 30.1 | 9.1 | 38.7 |
| EMR Merging | 47.0 | 67.6 | 67.8 |
| IP-Merging | **61.9** | **82.3** | **84.6** |

We also conduct experiments by merging larger models in appendix C.1, and merging multiple models in appendix C.2, demonstrating the effectiveness of our approach. Our method can further improve the performance of math MLLM, and is effective when merging larger or multiple models.

To validate that our approach does not harm other abilities in the MLLMs, we further conduct experiments on evaluating the models on the general QA benchmarks in table 3. Compared to the baseline model, our approach is able to maintain stable model performance across the general benchmarks, while other methods (e.g., Task Arithmetic and Ties Merging) decrease the performance. As our method only selects the crucial parameters related to math reasoning, our approach does not interfere with parameters that are important to other general abilities. Hence, our method improves the math reasoning abilities without interfering with the general abilities.

## 5.3 Merging Math MLLM with Math LLM

We conduct the experiments of merging fine-tuned math MLLM (TableLLaVA-1.5-7B [56], G-LLaVA-7B [8]) and the Tora model using our proposed merging method, validating the effectiveness of improving fine-tuned math MLLM further. G-LLaVA is obtained by further fine-tuning the LLaVA using geometry reasoning data. After merging the math LLM, the geometry reasoning ability is further enhanced, which is validated by the 2.8% and 1.5% improvement on GPS and FQA tasks. TableLLaVA is obtained by further fine-tuning the LLaVA using table reasoning data. By merging math reasoning LLM, our method can further improve the model's performance on table reasoning tasks such as FQA by 3.7%. VL-Rthinker-7B [35] is one recent reasoning model that was tuned from Qwen2.5-7B via reinforcement learning. By merg-

Table 4: We merge Math MLLM with the math LLM model. **Bold** represents the best performance. "Avg" is the average performance.

| Approach | MathVista | | | | | |
|---|---|---|---|---|---|---|
| | TQA | GPS | VQA | FQA | MWP | Avg |
| *G-LLaVA-7B as Base Model* | | | | | | |
| Base Model | 29.1 | 48.6 | **33.5** | 19.3 | 11.3 | 28.0 |
| +IP Merging | **32.9** | **51.4** | 31.8 | **20.8** | **12.9** | **29.6**[1.6↑] |
| *TableLLaVA-1.5-7B as Base Model* | | | | | | |
| Base Model | 34.2 | 27.4 | 29.6 | 24.9 | 41.9 | 30.9 |
| +IP Merging | **41.8** | **27.9** | **30.2** | **28.6** | **43.6** | **34.0**[3.1↑] |
| *VL-Rethinker-7B as the Base Model* | | | | | | |
| Base Model | 70.9 | 76.0 | 54.2 | **78.1** | 80.6 | 72.7 |
| +IP Merging | **71.5** | **79.3** | **58.7** | 77.7 | **86.0** | **75.2**[2.5↑] |

ing the Qwen2.5-Math model, the reasoning performance is further enhanced by 2.5 on MathVista, with noticeable improvement of 5.4% on the MWP subset. All of these demonstrate that our method can further improve the math reasoning abilities of both base MLLM and fine-tuned math MLLM.

## 5.4 Ablation Studies

We conduct the ablation experiments of the proposed method using the LLaVA model on MathVista in table 5. By selecting the parameters and applying our method individually, the performance is elevated by 3.3%, 3.7%, and 3.1%, respectively. More parameter selection analysis can be found in appendix D. Combining Parameter Selection and Projection increases accuracy to 26.3%, suggesting that projecting the selected parameters into the subspace provides better alignment. The combination of parameter selection and rescaling demonstrates that rescaling improves the utilization of selected parameters. The combination of rescaling and projection achieves 26.7%, highlighting the complementary benefits of parameter adjustment and alignment in improving per-

Table 5: Ablation results.

| Components | | | Acc. |
|---|---|---|---|
| Selection | Rescale | Projection | |
| ✓ | | | 25.8 |
| | ✓ | | 26.2 |
| | | ✓ | 25.6 |
| ✓ | | ✓ | 26.3 |
| ✓ | ✓ | | 26.3 |
| | ✓ | ✓ | 26.7 |
| ✓ | ✓ | ✓ | **28.2** |

formance. When all three components are applied together, the model achieves the highest accuracy. This improvement demonstrates the necessity of carefully selecting, rescaling, and aligning reasoning-related parameters.

## 5.5 Selected Parameter Analysis

We visualize the selected parameters for merging in fig. 5. Most of the selected layers are MLP components located in the middle and latter parts of the model, while only a few attention layers in the early stages are chosen. This pattern aligns with recent findings that knowledge and reasoning skills in LLMs are mainly encoded in deeper MLP layers [7, 3]. The selected layers in layers 17–31, and a few in layers 3–4 indicate that our merging primarily operates on high-level semantic representations while preserving early-layer perceptual alignment. This supports the view that reasoning transfer mainly benefits from modifying deeper MLP pathways rather than early attention dynamics. We further analyze selected layers in MLLM before and after math reasoning in appendix D.

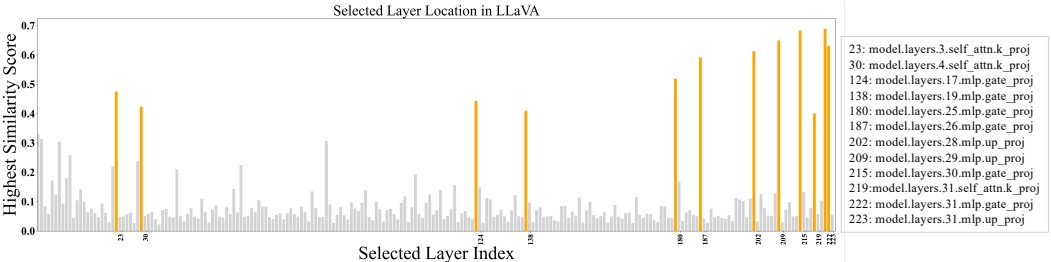

Figure 5: Selected layers in the LLaVA model for merging.

## 5.6 Case Analysis

We demonstrate two examples that the merged model improves reasoning ability compared to the base model in fig. 6. In the first example, the base model fails to read the table and outputs meaningless text, showing that it cannot handle structured data. In contrast, the merged model correctly reads the numbers, clearly explaining the rate of change. This shows that merging helps the model understand tables and perform basic numerical reasoning. In the second example, the task involves a geometry problem. The base model guesses the answer without any explanation, while the merged model reasons step by step. This reasoning is both correct and interpretable. See more cases in appendix E.

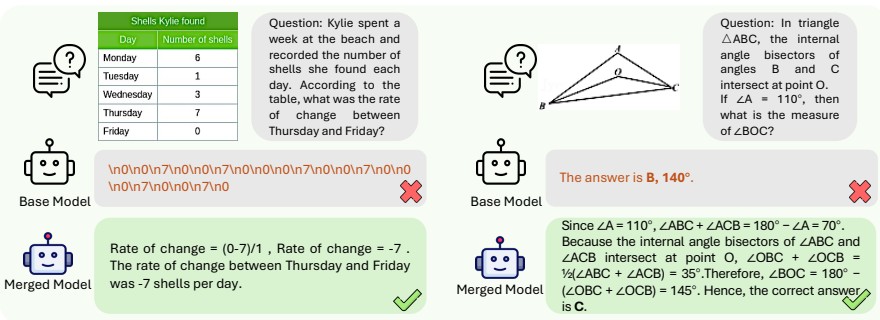

Figure 6: Case study of examples are correctly answered after merging the math LLM.

## 6 Conclusion

In this paper, we aim to enhance the math reasoning abilities of MLLMs by directly absorbing them from Math LLMs without tuning. However, it is challenging to merge models with different modalities due to the large gap in parameter spaces between models. To this end, we propose a model merging method, namely **IP-merging**, which mainly consists of two parts: i.e., parameter Selection and Projection. Parameter selection identifies crucial parameters associated with math reasoning, while parameter projection aligns the MLLM and math LLM by rescaling parameters and projecting the parameters into the subspace of the MLLM. The proposed approach is tune-free and efficient to implement. Experimental results demonstrate the IP-merging method can successfully enhance the math reasoning abilities of MLLMs without harming their other abilities. In the future, we will extend the method to merge models with different foundation LLM architectures.

**Limitation:** Our proposed method is effective for merging the MLLM and math LLM with the same size and sharing the same foundation LLM architecture (e.g., LLaMA-2). Otherwise, it is challenging to find those associated layers to merge.

## 7 Acknowledgments

The work was partially supported by the following: National Natural Science Foundation of China under No. 92370119, 62436009, 62276258 and 62376113.

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

## A    Border Impact

This work presents IP-Merging, a novel tuning-free method that effectively transfers mathematical reasoning capabilities from specialized math LLMs to multi-modal LLMs (MLLMs). Our approach does not pose any potential societal impacts or ethical concerns. We rely solely on publicly available models and datasets, none of which require specific acknowledgment.

## B    Detailed Experimental Settings

### B.1    Datasets

We test our models math reasoning benchmarks MathVista [23], MathVerse [53], DynaMath [59] and MathVision [36], general QA benchmarks MMMU [50], TextVQA [31] and MMBench [22].:

- **MathVista** assesses the MLLMs' multimodal mathematical skills, the testing data can be divided into five subsets: Figure Question Answering (FQA), Geometry Problem Solving (GPS), Math Word Problems (MWP), Textbook Question Answering (TQA), and Visual Question Answering (VQA). For evaluation, following [23, 30], we first employ GPT-4 to extract the predicted choices or answers from responses, then report the answer accuracy, which determines whether the final answer matches the ground truth.

- **MathVerse** includes a diverse set of math problems that require understanding and reasoning over both textual and visual information, such as charts, diagrams, and equations. The testing data can be divided into five subsets, i.e., Text Dominant, Text Lite, Vision Intensive, Vision Dominant and Vision Only.

- **DynaMath** is a dynamic visual mathematics benchmark developed for comprehensive evaluation of vision-language models. It consists of 501 carefully curated seed problems, each implemented as a Python program, enabling the assessment of a model's generalization capability by measuring its robustness across multiple input variations derived from the same underlying seed question.

- **MathVision** is a carefully constructed dataset comprising 3,040 high-quality visual mathematics problems collected from real-world math competitions. Covering 16 mathematical domains and organized into 5 difficulty levels, it offers a broad and balanced benchmark for assessing the visual mathematical reasoning capabilities of MLLMs.

- **MMMU** includes 900 evaluation samples and covers six core disciplines: Art & Design, Business, Science, Health & Medicine, Humanities & Social Science, and Technology & Engineering, making it suitable for assessing the general knowledge of MLLLM.

- **TextVQA** evaluates a model's ability to read and reason over textual information embedded within images. It requires integrating visual understanding with text recognition to answer questions grounded in image content. The dataset comprises 45,336 questions spanning 28,408 images sourced from the OpenImages dataset, providing a large-scale benchmark for visual-text reasoning.

- **MMBench** provides multiple-choice questions in both English and Chinese, facilitating direct and fair comparisons of MLLM performance across languages. Overall, it serves as a systematically constructed and objective benchmark for comprehensive and reliable evaluation of MLLM models. We use English dev set in the experiments.

### B.2    Models and Comparison Methods

We employ the LLaVA series [20], Qwen 2 series [37] and InternVL3 [58] series as our base model. We use other fine-tuned Math LLMs such as Tora series models, MetaMath models [9, 47] and Qwen2-Math models [42]. The pretrained foundation LLM of LLaVA models, Tora models and Metamath is the LLaMA-2 [34] model. The pretrained foundation of Qwen series models is the Qwen-2 model [37]. The InternVL model is trained based on the Qwen2.5 models. We use RTX 3090 GPUs for all of our experiments. We compare our proposed methods with prevailing model merging techniques:

- **Base Model** The performance of base MLLM on three tasks, we reproduce the results with the officially released code.

- **Task Arithmetic [15]** combines all the task vectors extracted from the models into one multi-task model.

- **Ties Merging [41]** addresses task inference by pruning redundant parameters. The process involves three steps: Trim, Elect Sign, and Disjoint Merge.

- **EMR Merging [14]** computes one unified task vector and computes task-specific masks based on a unified task vector. The final task vector is computed by combining all the masked task vectors and weighting all the task vectors by rescale parameters.

Table 6: List of MLLMs and LLMs.

| Models | Type | Pretrained Base Model | Source LLM |
|---|---|---|---|
| LLaVA-V1.5-7B | MLLM | Vincuna-v1.5-7B | Llama-2-7B |
| Table-LLaVA-V1.5-7B | MLLM | Vincuna-v1.5-7B | Llama-2-7B |
| LLaVA-Next-7B | MLLM | Vincuna-v1.5-7B | Llama-2-7B |
| LLaVA-V1.6-7B-Llama3-8B | MLLM | Llama-3-8B | Llama-3-8B |
| LLaVA-V1.5-13B | MLLM | Vincuna-v1.5-13B | Llama-2-13B |
| InternVL3-8B-Instruct | MLLM | Qwen2.5-7B | Qwen2.5-7B |
| VL-Rethinker | MLLM | Qwen2.5-7B | Qwen2.5-7B |
| Qwen2-VL-7B-Instruct | MLLM | Qwen2-7B | Qwen2-7B |
| Qwen2-Math-7B-base | LLM | Qwen2-7B | Qwen2-7B |
| Tora-7b | LLM | Llama-2-7B | Llama-2-7B |
| Tora-Code-7B | LLM | CodeLLaMA-7B | Llama-2-7B |
| Tora-Code-13B | LLM | CodeLLaMA-13B | Llama-2-7B |
| WizardMath-7B-V1.0 | LLM | Llama-2-7B | Llama-2-7B |
| Tora-7b | LLM | Llama-2-7B | Llama-2-7B |
| MetaMath-7B | LLM | Llama-2-7B | Llama-2-7B |
| OpenO1-Llama3-8B | LLM | Llama-3-8B | Llama-3-8B |
| DeepSeek-R1-distilled-Qwen-7B | LLM | Qwen2.5-Math-7B | Qwen2.5-7B |
| DeepSeek-R1-distilled-Llama-3-8B | LLM | Llama-3-8B | Llama-3-8B |

We list the models used in the experiments in table 6.

## B.3 Details of Hyperparameter Selection

Following previous works in model merging [41, 14], we adopt the grid search for hyperparameters for the baseline methods, specifically, we set the hyperparameters based on the following range:

- **Task Arithmetic [15]** involves the scaling coefficients for merged task vectors, which are set ranging from [0.1, 0.2, 0.3, 0.4, 0.5, 0.6, 0.7, 0.8, 0.9].

- **Ties Merging [41]** involves the scaling coefficient and ratio to retain large parameters, the scaling coefficients are set ranging from [0.1, 0.2, 0.3, 0.4, 0.5, 0.6, 0.7, 0.8, 0.9], ratio to retain parameters with largest-magnitude values: [0.1, 0.2, 0.3].

- **EMR Merging [14]** does not involve specific hyperparameters.

- **IP Merging** involves the similarity threshold to determine whether the layer should be selected for merging.

# C  Further Experiments

## C.1  Merge Larger Models

We conduct experiments on merging larger models in table 7. The experiments are conducted using models with 13B parameters. When dealing with larger models, our approach can also boost the performance compared to other model merging methods by 2.8% on average. In subtasks such as VQA, where math-related knowledge is highly demanded, our method achieves a 3.3% performance gain.

Table 7: Experiments of merging 13B models. We merge LLaVA-13B with Tora-code-13B using different merging methods. **Bold** represents the best performance. "Avg" is the average performance.

| Approach | MathVista | | | | | | |
|---|---|---|---|---|---|---|---|
| | Param | TQA | GPS | VQA | FQA | MWP | Avg |
| Base Model | 13B | 41.1 | 25.0 | 34.1 | 21.9 | 16.1 | 26.7 |
| Task Arithmetic | 13B | 39.9 | **34.6** | 29.1 | 22.3 | 7.0 | 26.0 |
| Ties Merging | 13B | 38.6 | 29.3 | 34.1 | 22.3 | 17.2 | 25.9 |
| **IP-Merging** | 13B | **42.4** | 26.0 | **37.4** | **23.1** | **18.8** | **28.5** +1.8 ↑ |

## C.2  Merge Multiple Models

We conduct experiments on emerging multiple models to verify the scalability of our method in table 8. We use LlaVA-Next-7B as the foundation MLLM, then merge multiple LLMs, such as the Tora Model and MetaMath-based models. As is shown in the table 8, by merging more math reasoning models, the performance of math reasoning MLLM can be further improved.

Table 8: Experiments of merging multiple models on MathVista.

| Models | MathVista | | | | | | | | |
|---|---|---|---|---|---|---|---|---|---|
| | Params | Approach | Merged LLM | TQA | GPS | VQA | FQA | MWP | Avg |
| MLLM | 7B | Base Model | None | 44.9 | 26.9 | **33.5** | 32.3 | 17.8 | 30.7 |
| MLLM+ 1 LLM | 7B | IP-Merging | Tora-code-7B | 46.2 | **32.2** | 30.2 | 33.8 | 18.3 | 31.9 +1.2 ↑ |
| MLLM+ 2 LLMs | 7B | IP-Merging | Tora-code-7B MetaMath-7B | **47.5** | 26.0 | 32.4 | **35.7** | **23.7** | **32.7** +2.0 ↑ |

## C.3  Merge Different System-1 LLMs and System-2 LLMs

By employing the proposed method, we conduct experiments on merging different reasoning pattern LLMs with base MLLM LLaVA-7B in table 9. We compare system-1 thinking LLMs such as WizardMath, MetaMath, Tora and Tora-code models. Merging Tora-code yields the best performance. Different from other models, Tora-code uses the high-quality reasoning data involved with the critique process. We believe Tora outperforms others for two main reasons: (1) CoT & PoT Collaboration: Tora employs a collaborative reasoning approach that integrates CoT and PoT to solve problems. In contrast, WizardMath and MetaMath rely solely on CoT. (2) Reflection Mechanism: Tora's training data incorporates a reflection-based correction process, where incorrect responses are analyzed and revised, contributing to improved reasoning abilities. Others merely filter incorrect reasoning data. By further fine-tuning the code llama, Tora-code is able to perform code-like reasoning on math problems, which exhibits strong performance compared to other math LLM on text-based math reasoning datasets such as GSM8K and Math. This experiment also reveals one interesting observation: math reasoning abilities obtained by Tora series models are more transferable to MLLM. We further conduct experiments on merging system-2 thinking LLMs such as OpenO1 [27] and the DeepSeek-R1-distilled-LLaMA3 model [10], demonstrating the effectiveness of our method. Our method further improves the model performance by merging the long CoT LLMs, obtaining a 2.3% performance gain on average.

Table 9: Experiments of merging different System-1 LLMs and System-2 LLMs on MathVista.

| MathLLM | MathVista | | | | | | |
|---|---|---|---|---|---|---|---|
| | Params | TQA | GPS | VQA | FQA | MWP | Avg |
| *LLaVA-1.5-7B with System-1 LLMs* | | | | | | | |
| Base Model | 7B | 36.1 | 22.1 | 37.4 | 20.8 | 14.0 | 25.2 |
| MetaMath-V1.0-7B | 7B | 36.1 | 22.6 | 34.1 | 23.4 | 15.6 | 25.7 +0.5 ↑ |
| WizardMath-7B | 7B | 41.4 | 25.0 | 34.1 | 23.4 | 13.4 | 26.6 +1.4 ↑ |
| Tora-V1.0-7B | 7B | 39.9 | **26.9** | 34.6 | **26.7** | 12.9 | 27.4 +2.2 ↑ |
| Tora-code-V1.0-7B | 7B | **43.7** | 21.6 | **40.8** | 24.9 | **15.1** | **28.2** +3.0 ↑ |
| *LLaVA-1.6-LLaMA3-8B with System-2 LLMs* | | | | | | | |
| Base Model | 8B | 50.6 | 29.3 | 38.6 | 46.5 | 23.1 | 37.8 |
| OpenO1-LLaMA3-8B | 8B | 51.9 | 29.8 | 40.2 | **46.5** | 24.7 | 38.7 +0.9 ↑ |
| DeepSeek-R1-distilled-LLaMA3 | 8B | **51.9** | **32.7** | **40.8** | 45.7 | **29.6** | **40.1** +2.3 ↑ |

## C.4 Similarity Threshold and Time Cost Analysis

We compare the time cost of different model merging algorithms in fig. 7(a). We conduct this experiment of merging 7B MLLM and one math reasoning LLM on a GPU server with 8-card Nvidia RTX 3090. As task arithmetic does not involve operations that modify the models, this method takes the least time. Ties merging and EMR merging involve comparing the sign of each parameter. Thus, they go through all parameters, which can be time-consuming. Since our method directly operates parameters of each layer (formatting as matrix), our method can be accelerated by GPUs and takes less time to merge the models.

We show the results of different similarity thresholds in fig. 7(b), see more results in ap-

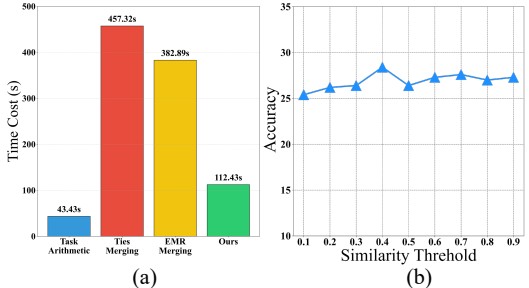

(a)                    (b)

Figure 7: (a) Time cost of our method compared to other methods. (b) The results of different similarity thresholds.

pendix C.5. The performance fluctuates within a small range with higher similarity. We further compare the time cost of different model merging algorithms in fig. 7(a). Task arithmetic is the fastest since it doesn't change the models. Ties and EMR merging are slower because they process all parameters. Our method is faster as it works on each layer's parameters and can be accelerated by GPUs and takes less time to merge the models.

## C.5 Results of Hyperparameter Experiments

We conduct ablation experiments for the hyperparameters in table 10. We show the results of different scaling coefficients in the task vector, the proportion of the retained parameters and the scaling coefficients in ties merging, and similarity thresholds (i.e., $S_\alpha$ in eq. (6)) in our method. We can see that for the Llava models, the threshold around 0.3 and 0.4 provides the best performance, balancing the number of layers associated with math reasoning to be merged and the importance of the layers to math reasoning. For the Qwen model, 0.6 is the optimal choice. We can also see that with higher thresholds, the performance fluctuates within a small range. We also show the comparative methods of ties merging and task vector, the performance is sensitive to the different scaling coefficients or the rate of the retained parameters.

Table 10: Results of different hyperparameters. **Bold** represents the best performance.

| Methods | LLaVA-1.5-7B | | | Qwen-2-VL | | |
|---|---|---|---|---|---|---|
| | **MathVista** | **MathVerse** | **MMMU** | **MathVista** | **MathVerse** | **MMMU** |
| **Base Model** | 25.2 | 11.3 | 34.2 | 55.4 | 24.8 | 50.7 |
| **Task Vector** | | | | | | |
| scale=0.1 | 21.0 | 6.5 | 24.4 | 25.8 | 6.0 | 29.3 |
| scale=0.2 | 25.2 | 7.9 | 25.6 | 27.8 | 5.9 | 32.6 |
| scale=0.3 | 22.2 | 8.5 | 26.6 | 30.8 | 8.8 | 29.1 |
| scale=0.4 | 24.5 | 3.6 | 26.3 | **31.1** | **9.3** | **31.0** |
| scale=0.5 | 23.1 | 4.4 | **30.2** | 28.9 | 7.6 | 26.0 |
| scale=0.6 | **24.8** | 5.4 | 25.8 | 28.1 | 4.9 | 26.9 |
| scale=0.7 | 23.3 | 6.2 | 26.2 | 28.5 | 1.4 | 24.9 |
| scale=0.8 | 23.9 | 8.0 | 24.4 | 29.2 | 0.0 | 25.2 |
| scale=0.9 | 20.9 | **9.8** | 25.6 | 24.1 | 0.0 | 23.9 |
| **Ties Merging** | | | | | | |
| retain=0.3,scale=0.1 | 22.9 | 5.9 | 25.8 | 24.9 | 1.5 | 27.6 |
| retain=0.3,scale=0.2 | 22.9 | 4.8 | 24.2 | 26.5 | 1.2 | 27.0 |
| retain=0.3,scale=0.3 | 24.8 | 7.3 | 25.8 | 26.2 | 2.3 | 27.2 |
| retain=0.3,scale=0.4 | 23.1 | 6.9 | 25.6 | 27.9 | 0.6 | 26.4 |
| retain=0.3,scale=0.5 | 26.1 | 1.9 | 22.6 | 27.6 | 0.6 | 26.2 |
| retain=0.3,scale=0.6 | 25.7 | 1.8 | 26.2 | 26.7 | 0.4 | 27.1 |
| retain=0.3,scale=0.7 | 26.6 | 0.0 | 25.7 | 26.8 | 0.1 | 25.7 |
| retain=0.3,scale=0.8 | 26.6 | 0.0 | 22.6 | 27.0 | 0.0 | 24.8 |
| retain=0.3,scale=0.9 | 25.0 | 0.0 | 25.8 | 24.6 | 0.0 | 26.2 |
| retain=0.2,scale=0.1 | 23.6 | 6.4 | 25.3 | 26.3 | 2.1 | 28.1 |
| retain=0.2,scale=0.2 | 23.6 | 4.6 | 24.0 | 29.2 | 2.3 | 26.0 |
| retain=0.2,scale=0.3 | 24.1 | 5.3 | 26.0 | 28.5 | 1.9 | 29.2 |
| retain=0.2,scale=0.4 | 25.6 | 7.1 | 26.0 | 27.9 | 1.2 | 26.9 |
| retain=0.2,scale=0.5 | 26.1 | 0.5 | 21.0 | 27.5 | 3.4 | 25.2 |
| retain=0.2,scale=0.6 | 26.8 | 0.0 | **27.3** | 27.2 | 0.3 | 23.9 |
| retain=0.2,scale=0.7 | 24.9 | 0.0 | 21.4 | 25.0 | 0.3 | 23.1 |
| retain=0.2,scale=0.8 | 24.8 | 0.0 | 24.1 | 24.8 | 0.0 | 28.3 |
| retain=0.2,scale=0.9 | 23.3 | 0.0 | 23.9 | 24.8 | 0.0 | 25.8 |
| retain=0.1,scale=0.1 | 23.2 | 6.2 | 25.7 | 27.5 | 1.5 | 30.0 |
| retain=0.1,scale=0.2 | 22.9 | 5.7 | 25.0 | 27.5 | 3.5 | 30.6 |
| retain=0.1,scale=0.3 | 23.6 | 6.4 | 24.4 | 27.5 | 3.4 | **30.8** |
| retain=0.1,scale=0.4 | 22.6 | **10.7** | 23.4 | **29.3** | **5.3** | 28.8 |
| retain=0.1,scale=0.5 | 24.3 | 6.7 | 25.0 | 27.0 | 2.9 | 27.3 |
| retain=0.1,scale=0.6 | 24.8 | 0.3 | 24.3 | 25.0 | 0.9 | 27.3 |
| retain=0.1,scale=0.7 | **27.1** | 0.0 | 26.9 | 26.3 | 0.5 | 24.4 |
| retain=0.1,scale=0.8 | 27.1 | 0.0 | 20.4 | 25.8 | 0.3 | 27.1 |
| retain=0.1,scale=0.9 | 24.8 | 0.0 | 25.8 | 26.1 | 0.2 | 29.1 |
| **EMR Merging** | 25.0 | 10.4 | 34.8 | 40.8 | 17.6 | 41.8 |
| **IP Merging** | | | | | | |
| Sim threshold=0.1 | 25.4 | 14.5 | 34.0 | 59.3 | 27.8 | 50.2 |
| Sim threshold=0.2 | 26.2 | 14.8 | 34.2 | 59.7 | 28.0 | 50.2 |
| Sim threshold=0.3 | 26.4 | **15.3** | **34.4** | 59.8 | 27.9 | 49.8 |
| Sim threshold=0.4 | **28.4** | 14.7 | 33.9 | 59.7 | 28.0 | 49.8 |
| Sim threshold=0.5 | 26.4 | 14.9 | 34.2 | 59.7 | 27.9 | 49.8 |
| Sim threshold=0.6 | 27.3 | 14.6 | 34.2 | **60.2** | **28.5** | **50.7** |
| Sim threshold=0.7 | 27.6 | 14.4 | 34.2 | 60.1 | 28.5 | 50.7 |
| Sim threshold=0.8 | 27.0 | 14.4 | 34.2 | 60.1 | 28.4 | 50.7 |
| Sim threshold=0.9 | 27.3 | 14.3 | 34.2 | 60.1 | 28.4 | 50.7 |

# D   Analysis of Selected Parameters

We visualize the proportion and composition of the selected parameters in fig. 8. As shown in the figure, the selected layers account for less than 10% of the total model parameters, with the majority concentrated in the MLP layers. This observation aligns with recent studies on knowledge storage in LLMs, which suggest that most knowledge and skills are encoded within the MLP layers [7, 52]. Since Table-LLaVA is fine-tuned on math reasoning datasets, it has already acquired a certain level of mathematical reasoning ability. Consequently, our selection process identifies a higher proportion of reasoning-related layers in Table-LLaVA compared to the base model, LLaVA. To further analyze the distribution of these selected layers, we plot their locations in fig. 8(c) and (d). The visualization reveals that most reasoning-associated layers are concentrated in the latter part of the model, suggesting that deeper layers play a crucial role in encoding mathematical reasoning skills.

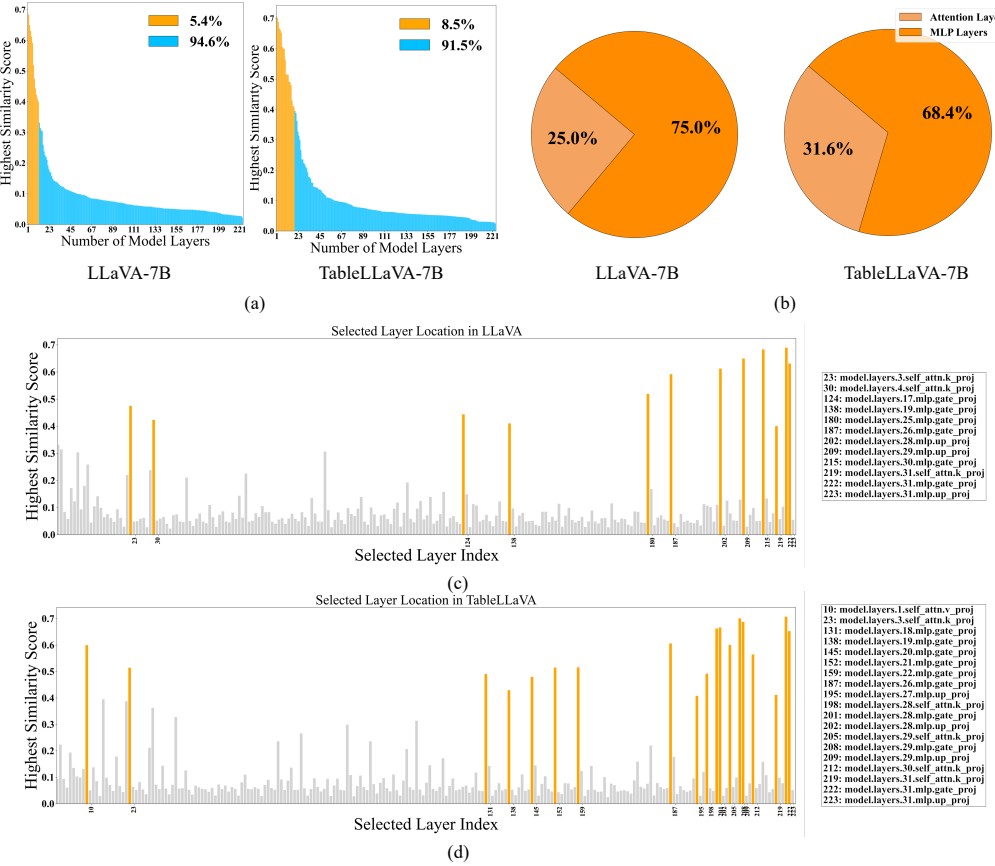

Figure 8: (a) The proportion of selected layer in two MLLMs. (b) The proportion of attention layers and MLP layers in selected layers.(c) Selected layers' location in LLaVA. (d) Selected layers' location in Table-LLaVA.

# E    User Case Analysis

As illustrated in fig. 9, the merged model effectively solved a symbolic reasoning problem by following logical steps to infer intermediate relations and reach the correct answer, showing that mathematical reasoning knowledge from the text-based LLM was successfully transferred and integrated into the multimodal model. This demonstrates that our merging strategy not only enhances reasoning depth but also improves consistency across steps, leading to more interpretable and reliable outputs. In contrast, the failure case occurred in a geometric reasoning scenario where the merged model predicted an incorrect angle despite the visual clues being straightforward. This reveals that while the merged model excels at structured symbolic inference, it still struggles to align textual reasoning with precise spatial understanding.

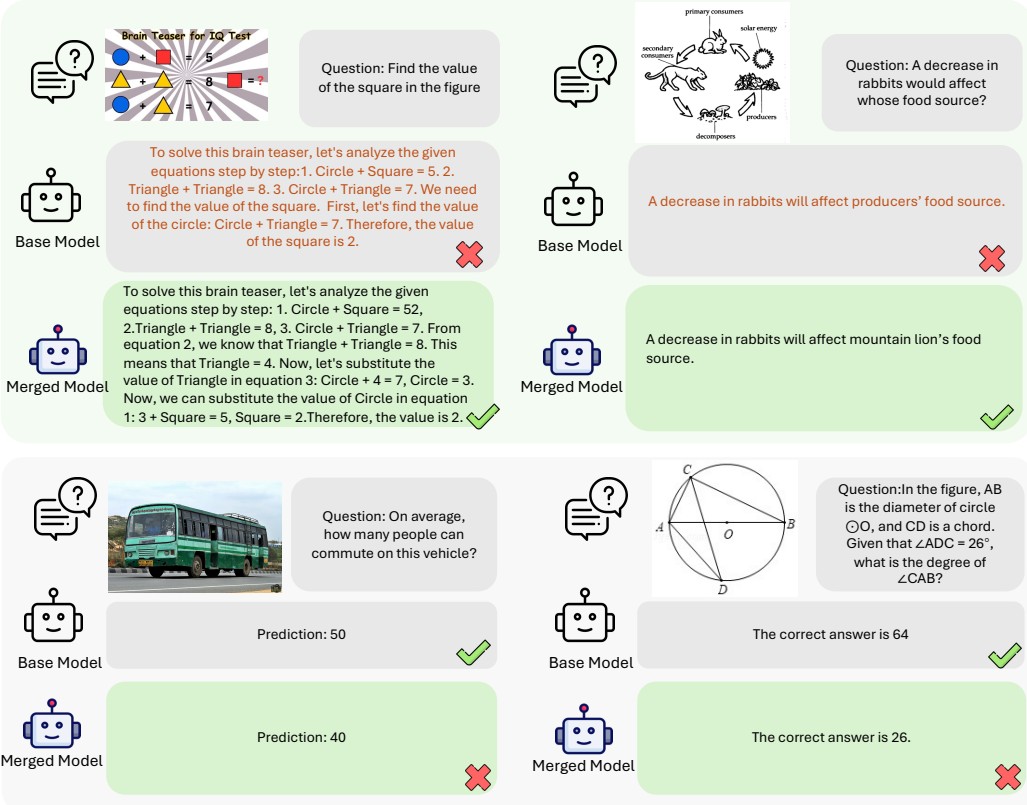

Figure 9: Successful cases and failure cases studies.

## F   Method Overview

IP merging firstly identifies key parameters in both the MLLM and the math LLM. It then projects the rescaled, selected parameters from the LLM into the subspace of the MLLM to achieve better alignment. Finally, the aligned parameters are merged into the MLLM. During the parameter identification phase, reasoning-related parameters are selected based on their similarity within a shared subspace. In the projection phase, these parameters are rescaled and aligned to minimize the discrepancy between the two models. The complete procedure is illustrated in fig. 10. We visualize the process of IP-Merging as follows:

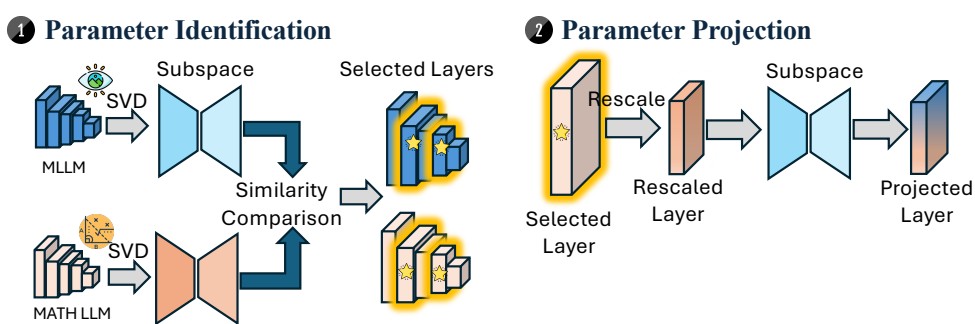

Figure 10: The general process of IP merging.

---

**Algorithm 1** IP-Merging
___

**Input:** Parameters of MLLM $\boldsymbol{W}_{MLLM}$ and Math LLM $\boldsymbol{W}_{Math}$, Pretrained Model $\boldsymbol{W}_0$; Threshold $S_\alpha$; Number of layers $N$.
**Output:** Math reasoning model $\boldsymbol{W}_{MathMLLM}$.
Compute task vectors: $\Delta\boldsymbol{W}_{MLLM}$ and $\Delta\boldsymbol{W}_{Math}$.
**for** $n = 1$ **to** $N$ **do**
    Compute SVD decomposition of $\Delta\boldsymbol{W}_{MLLM}^n$ and $\Delta\boldsymbol{W}_{Math}^n$ using Equation 3.
    Compute the similarity scores $\{S_1^n, S_2^n, \ldots, S_d^n\}$ for the $n$-th layer using definition 1.
    **if** $S_1^n > S_\alpha$ **then**
        Compute the scaling factor $\lambda_n$ using Equation 7.
        Compute the importance score $\gamma_n$ using Equation 8.
        Project $\Delta\boldsymbol{W}_{Math}^n$ into the subspace to obtain $\Delta\boldsymbol{W}_{Math-P}^n$ using Equation 9.
        **return** $\boldsymbol{W}_0^n + \Delta\boldsymbol{W}_{MLLM}^n + \Delta\boldsymbol{W}_{Math-P}^n$
    **else**
        **return** $\boldsymbol{W}_0^n + \Delta\boldsymbol{W}_{MLLM}^n$
    **end if**
**end for**
___

