# OpenReview forum: "Can MLLMs Absorb Math Reasoning Abilities from LLMs as Free Lunch?"
_NeurIPS.cc/2025/Conference — NeurIPS 2025 poster_

### Official Review · Reviewer_hd9w · 2025-06-30

**Clarity:** 3
**Significance:** 2
**Originality:** 3
**Rating:** 4
**Confidence:** 3

**Summary:**

This paper proposes IP-Merging, a tuning-free model merging technique to enhance the mathematical reasoning capabilities of MLLMs by merging them with math-specialized LLMs. It first identifies math-reasoning-associated parameters, rescales and projects them into a shared subspace, and merges the models. Compared with other model merging baselines, IP-merging is shown to more effectively improve math reasoning performance on MathVerse and MathVista without degrading general capabilities on MMMU.

**Questions:**

Please refer to the weaknesses section first.

1. Can this method be applied to domains beyond math? It seems such a merging strategy could be generalizable to other domains—could you elaborate? If so, why only focus on math reasoning in this paper?

2. Could you provide qualitative examples where the merged MLLM successfully solves a problem that the base model fails to address? This would offer intuition and insight about the improvements.

**Ethical Concerns:**

["NO or VERY MINOR ethics concerns only"]

**Final Justification:**

Overall, I believe that the article is generally a positive and innovative attempt. The author's additional experiments, to some extent, show that IP-Merge has not harmed the performance on TextVQA, so I give it a positive evaluation.

Considering the limitations in the model merging scenario and the fact that the experimental data is still not sufficient to completely eliminate concerns about the free lunch claims, I maintain a borderline recommendation.

**Limitations:**

Yes

**Quality:**

3

**Strengths And Weaknesses:**

### **Strengths**

1. The paper provides a rigorous empirical analysis of why conventional merging methods are ineffective, convincingly demonstrating the "parameter space gap." This analysis offers a strong motivation for the proposed work.
2. The proposed IP-Merging method is novel and effectively addresses the diagnosed issues. Its training-free nature makes it a practical solution for enhancing model capabilities.
3. Experimental results effectively demonstrate that IP-Merging enhances the mathematical reasoning of MLLMs without compromising performance on MMMU, and IP-Merging is shown more competitive performance over baselines.
4. Overall, the paper is well-written, clear, and easy to follow. (Though I still have some suggestions for improvement, please see below.)

### **Weaknesses**

1. As acknowledged in the limitations section, the method currently only supports merging models of the same size and base architecture. This indeed significantly limits its applicability and potential adoption.

2. The "free lunch" claim is not sufficiently supported—evaluating only on MMMU is not enough. A broader set of evaluations is needed to verify whether the merged models truly retain their original capabilities.

3. Beyond performance metrics, an analysis of error or failure cases before and after merging could offer additional insights and perspectives.

4. Ablation studies on the hyperparameters $\alpha_1$ and $\alpha_2$ are missing.



### **Writing & Presentation (Minor)**

1. The experiments involving the merging of multiple models are valuable. I recommend moving them from the appendix into the main text and providing a more in-depth analysis.
2. The readability of Table 9 is poor. Consider replacing it with line charts (e.g., a matrix of subfigures), which could illustrate trends more effectively.

---

> ### Author Rebuttal · Authors · 2025-07-31
>
> We are thankful for your valuable comments. Here, we address your concerns and questions point by point.
>
> - **W1: Limitations of merging different structures**
>
> As noted in the paper’s limitations, our method is designed for merging MLLMs and math LLMs that share the same base architecture and size. This ensures we can reliably identify aligned layers for merging. In practice, we believe this is not a major limitation, as most MLLMs are built on top of their corresponding LLMs. For example, LLaVA is based on LLaMA-based models, and Qwen2-VL on Qwen2-based models, making such model pairs common. **Currently**, **this architectural constraint is shared by most existing training-free merging methods**. which typically assumes the models have the same size and cater to the same modality. Our work goes a step further by enabling cross-modality merging to improve multi-modal reasoning.
>
> - **W2: More general benchmark results**
>
> We are grateful for your comments. Following your suggestion, **we conduct extra experiments on TextVQA to support our claim.** These results are shown below. Our method allows the models to maintain their general capabilities.
>
> | Model | **TextVQA** |
> | --- | --- |
> | LLava-1.5-7b | 47.50 |
> | +IP merging | **47.52** |
> | Qwen2-vl-7b-instruct | 83.8 |
> | +IP merging | **84.0** |
> - **W3&Q2:  Failure / Successful case study**
>
> Thank you for the valuable advice. We illustrate two cases in the MathVista below for your reference. As shown in the Failure Case, the merged model may fail on problems where visual common-sense reasoning is required. This suggests that merging may impair the model’s ability to reason over visual information and commonsense knowledge, potentially caused by math-focused weights after merging.  As shown in the successful case, the merged model is better in cases where statistical reasoning and arithmetic reasoning are demanded. In the revision, we will add one section of case study to show more qualitative results and analysis and provide a more intuitive understanding.
>
> | Cases | Question | Comments |
> | --- | --- | --- |
> | **Faliure Case:** "MathVista-id:194","image":"images/194.jpg" | On average, how many people can commute on this vehicle? | **Note: This question requires the model to estimate the number of people that can be accommodated in a big green bus based on commonsense knowledge.** |
> | Base model(LLava): | Prediction: 50 **(True)** |  |
> | Merged model: | Prediction: 40 **(False)** | The model fail to do the commonsense reasoning of the people in a big bus. |
> | **Successful Case:** "MathVista-id:359,"image":"images/359.jpg"  | Kylie spent a week at the beach and recorded the number of shells she found each day. According to the table, what was the rate of change between Thursday and Friday? (Unit: shells per day) | **This question requires the model to look at a statistical table and calculate the change rate of the item.** |
> | Base model(LLava): | Prediction: 7\n0\n0\n7\n0\n0\n7\n0 **(False)**  |  |
> | Merged model: | Prediction: Rate of change = $\frac{0 - 7}{1}$, ****Rate of change = $-7$. The rate of change between Thursday and Friday was -7 shells per day.  **(True)** | The model reads statistics in the paper, and calculate the results of change rate.  |
> - **W4:  Scaling parameter α1 and α2**
>
> Sorry for the confusion.  **The scaling factors α₁ and α₂ are not hyperparameters**. As mentioned in line 296 of the main paper, α₁ is set to 1, and α₂  is computed based on the matrix norm using Eq. (7). **In our paper, the main hyperparameter is the similarity threshold, which we provide in Table 9 along with the hyperparameter results in the baseline.
>
> - **Wr&Pre:  Rearrange the experiments and optimize Tab 9**
>
> Thank you for the valuable suggestion. We will move the results of merging multiple models to the main paper and plot line figures to better showcase the results in the revision.
>
> - **Q1: Apply to other domains**
>
> Thank you for the thoughtful question. Math reasoning has been one crucial ability of LLMs and MLLM, hence we focus on math reasoning in our paper.  Essentially, our method relies on using a specialized model and its task vector to transfer capabilities from LLMs to corresponding MLLMs.  This setup is not restricted to math reasoning.  While prior works have explored model merging in text-only domains, our work extends this to the more challenging multi-modal setting. Our approach could potentially be applied to other domains where paired specific LLM and MLLM models exist. In case that more benchmarks and corresponding specialized models are available, we are happy to investigate extending our method in these domains.

---

> > ### Comment · Reviewer_hd9w · 2025-08-04
> >
> > I thank the authors for their detailed clarification.
> >
> > While I still have some concerns that restricting support to merging models of the same size and base architecture may limit the method’s applicability and broader adoption, I acknowledge that this is a pragmatic approach given the current stage of research.
> >
> > Additionally, the case study and corresponding insights are interesting. Overall, I believe the authors have addressed most of my concerns effectively with clarification, additional experimental results, and case studies.

---

> > > ### Author Response · Authors · 2025-08-07
> > >
> > > Thank you very much for your thoughtful feedbacks We truly appreciate your constructive comments and are glad that our clarifications and additional results helped address your concerns.

---

### Official Review · Reviewer_XRHa · 2025-07-02

**Clarity:** 2
**Significance:** 2
**Originality:** 2
**Rating:** 4
**Confidence:** 3

**Summary:**

This paper presents a method for improving the math reasoning capabilities of multimodal large language models (MLLMs) by directly merging parameters from math-focused LLMs, without additional training or data. The proposed IP-Merging approach identifies reasoning-related layers in both models and projects selected parameters from the math LLM into the MLLM’s subspace to address parameter misalignment. Experimental results on benchmarks such as MathVista and MathVerse, using models like LLaVA and Qwen2-VL, show that IP-Merging enhances math reasoning performance of MLLMs while maintaining their general abilities.

**Questions:**

**See W1&W2&W3&W4.** (The current evaluation mainly focuses on relatively small and outdated models. Can the authors provide results on more recent and larger MLLMs, especially those with stronger baseline performance? As plug-and-play methods often become less effective as baselines improve, it would be important to discuss whether IP-Merging still offers meaningful gains in these settings. If the improvements are marginal or absent with recent models, what are the main factors limiting its effectiveness?)

**Ethical Concerns:**

["NO or VERY MINOR ethics concerns only"]

**Final Justification:**

I appreciate the authors' efforts, and they have addressed my concerns during the rebuttal period.

**Limitations:**

yes

**Quality:**

3

**Strengths And Weaknesses:**

***Strengths***
1. The paper introduces IP-Merging, a new parameter merging method that enables MLLMs to directly acquire math reasoning abilities from math-specialized LLMs without any additional training or data.
2. The approach incorporates subspace similarity-based parameter selection and projection, effectively identifying and aligning reasoning-associated parameters between models to facilitate knowledge transfer.
3. Experimental results demonstrate that IP-Merging consistently improves the math reasoning performance of MLLMs on benchmarks such as MathVista and MathVerse, while maintaining the general capabilities of the models.

***Weaknesses***
1. **Lack of evaluation on recent reasoning models:** Although the paper reports results on G-LLaVA and similar math-specific models, these are based on older architectures (2023). For example, after applying IP-Merging, G-LLaVA only achieves a MathVista score of 29.6, whereas recent 7B reasoning models can reach nearly 70 on the same benchmark. The lack of evaluation on recent reasoning models makes it difficult to judge whether the effectiveness of IP-Merging generalizes to current high-performing models.
2. **Lack of diverse and strong general baselines:** The experiments only consider LLaVA-1.5-7B and Qwen-2-VL-7B as general MLLMs. The absence of more recent, larger, or more competitive models, as well as different training strategies and sizes, limits the assessment of IP-Merging’s robustness and general applicability.
3. **Insufficient benchmark coverage:** The reasoning benchmarks used in the experiments are limited to MathVista and MathVerse. Results on additional benchmarks such as MathVision, WeMath and Dynamath would provide a more comprehensive assessment. In particular, WeMath and Dynamath can highlight reasoning mechanism deficiencies, while MathVision focuses on more challenging problems.
4. **Experimental results and reporting require clarification:** There are inconsistencies in reported numbers, such as a MathVista score of 55.4 in the paper compared to the official 58.2. Additionally, for MMMU (Table 3), the baseline (Qwen-2-VL-7B) and IP-Merging results (including sub-categories) are exactly the same. These points require clarification, as they directly impact the credibility of the experimental conclusions.
5. **Architectural restriction:** IP-Merging requires both the MLLM and the math LLM to have the same architecture and parameter size, significantly limiting its applicability in practical scenarios involving heterogeneous or larger-scale models.

---

> ### Author Rebuttal · Authors · 2025-07-31
>
> We are thankful for the valuable comments. Here, we address your concerns and questions point by point.
>
> - **W1: Experiments of recent reasoning models**
>
> Thanks for the suggestions. Due to the limited rebuttal window, we conduct experiments on one recent SOTA reasoning MLLM VL-Rehinker-7B [1], which still demonstrates the effectiveness of IP-Merging in enhancing the performance of reasoning MLLM on various math reasoning benchmarks.  We are happy to report results on more reasoning models in our final version.
>
> | Model | **MathVista** | **MathVerse** | **MathVision** | **WeMath** | **MMMU_val** |
> | --- | --- | --- | --- | --- | --- |
> | VL-Rethinker-7B | 72.7 | 51.9 | 28.5 | 37.6 | 59.4 |
> | + IP merging | **75.2** | **52.4** | **29.6** | **39.4** | **60.4** |
> - **W2&W3: Diverse baselines and benchmarks**
>
> Following your kind suggestions, we further conduct experiments using Qwen2.5-VL-7B-Instruct,  InternVL3-8B-Instruct, and extend the Math-Benchmarks to MathVision, DynaMath and WeMath. As shown in the table, our merged model can further improve the math reasoning performance across different benchmarks.
>
> | Model | **Mathverse** | **MathVista** | **MathVision** | **Dynamath** | **Wemath** |
> | --- | --- | --- | --- | --- | --- |
> | Qwen2-VL-7B-Instruct | 24.8 | 55.4 | 17.2 | 40.8 | 23.6 |
> | +IP Merging | **28.5** | **60.2** | **19.1** | **41.0** | **24.4** |
> | llava-1.5-7B | 11.3 | 25.2 | 11.1 | 15.4 | 6.5 |
> | +IP Merging | **15.4** | **28.4** | **11.8** | **16.1** | **7.2** |
> | Qwen2.5 VL-7B-Instruct | 46.6 | 68.0 | 24.7 | 51.9 | 31.1 |
> | +IP merging | **46.8** | **69.7** | **26.3** | **52.5** | **33.5** |
> | InterVL3-8B-Instruct | 38.8 | 66.1 | 24.9 | 50.7 | 31.6 |
> | +IP meging | **39.0** | **67.6** | **25.2** | **51.4** | **33.0** |
> - **W4: Clarification on the experimental results**
>
> We are sorry for the confusion.  **(1) Inconsistencies in reported numbers on MathVista: For fair evaluation, we reproduce all the results using the public official evaluation code, parameters and prompts [2].** We also adopt the same setting in our merged models for fair evaluation. We applied the same setup to our merged models. Variations in hardware may also affect performance. For instance, we observed a score of 54.7 on an A800 GPU versus 55.4 on an RTX 3090. To ensure consistency, we used RTX 3090 and the same runtime environment for all experiments.
>
> **(2)** **Clarification for the same results on MMMU**: We are aware that the MMMU subsets have the same score. However, there do exist differences if we look into the statistics of each subset. For example, in Humanities and Social Science subsets of MMMU as shown below, we find that they are different in two subsets, including History and Sociology, although their sum is the same.
>
> **Base MLLM**: 'Overall-Humanities and Social Science': {'num': 120, 'acc': 66.667}, **'History': {'num': 30, 'acc': 70.0}**, 'Literature': {'num': 30, 'acc': 86.667}, **'Sociology': {'num': 30, 'acc': 56.667},** 'Psychology': {'num': 30, 'acc': 53.333}
>
> **Merged Model:** Overall-Humanities and Social Science': {'num': 120, 'acc': 66.667}, **'History': {'num': 30, 'acc': 66.667}**, 'Literature': {'num': 30, 'acc': 86.667}, **'Sociology': {'num': 30, 'acc': 60.0},** 'Psychology': {'num': 30, 'acc': 53.333}
>
> We suspect this might be due to the limited evaluation samples (900 samples). For your reference, we add experiments of TextVQA (5000 samples), where the base model achieves a score of 83.8 while the merged model achieves 84.0.  In summary, we are confident that our results are reproducible, and all reviewers are welcome to run the code in the anonymous link to verify our results.
>
> - **W5: Limitation on architecture**
>
> As noted in the limitations of our paper, our method is designed for merging MLLMs and math LLMs that share the same base architecture and size (e.g., LLaMA-2). This is necessary to reliably identify aligned layers for merging. Nonetheless, we believe this does not pose a major constraint in practice, as **Most existing MLLMs are built on top of their corresponding LLMs.** For example, LLaVA is based on LLaMA models, and Qwen2-VL is based on Qwen2 models, making such model pairs common in real-world scenarios. Currently, **this architectural constraint is a common limitation across train-free model merging methods,** which typically assume the same size and modality. Our work takes a step further by addressing cross-modality merging to enhance multi-modal reasoning capabilities.
>
> - **Questions:**
>
> We provide extra experimental results of more recent baseline models (e.g., LLaVA and Qwen series, Fine-Tuned or RL-based training strategy) and benchmarks (e.g., math-related and general benchmarks), demonstrating the effectiveness of our approach even on stronger baseline models.
>
> [1] Wang et.al, VL-Rethinker: Incentivizing Self-Reflection of Vision-Language Models with Reinforcement Learning, 2025
>
> [2] Wang et.al, Qwen2-VL: Enhancing Vision-Language Model's Perception of the World at Any Resolution, 2024

---

> > ### Comment · Reviewer_XRHa · 2025-08-05
> >
> > Dear Authors,
> >
> > Thank you for your clarification. Your response has addressed some of my concerns, and I am considering raising my score to 3. However, I still have several reservations regarding the experimental details. Overall, there are some abnormal results and settings in the experimental section. In particular, some of the reported improvements are within or even smaller than the margin of testing error, which requires further clarification.
> >
> > **1. On the consistency of MMMU scores:**
> > The explanation for the identical MMMU scores is not convincing. If the overall category results are identical while the subcategory results differ, I suggest that all fine-grained results should be presented. In the MMMU table, there are no identical scores, yet the two rows mentioned in the response are exactly the same, which is unusual and should be carefully analyzed in the paper.
> >
> > **2. On the effectiveness of IP merging:**
> > The additional results provided do not show any clear pattern. For instance, for an earlier model like LLaVA-1.5, one would expect greater improvements on certain benchmarks, but some benchmarks show gains of only 0.7, while newer models exhibit increases of 1.6. Please clarify whether you have identified any underlying patterns in these results.
> >
> > **3. On reproducibility:**
> > All tests were conducted on 3090 GPUs, and you mentioned that the results are better than those on A800, but still 3% lower than the official results. Please clarify why this is the case (3090 > A800). Is it due to hardware differences or other factors? If there are special settings or reproducibility issues, these should be clearly explained, ***especially since our own tests on A800 can match the official scores. (58.2%)***
> >
> > I hope the authors can further clarify these issues to enhance the credibility of the work. In particular, the first and third points are critical, as the associated errors are already greater than the reported experimental improvements.

---

> > > ### Author Response · Authors · 2025-08-07
> > >
> > > Thank you for your thoughtful suggestions. While we are confident these results are reproducible and credible,  we will address your further questions and concerns point by point as below.
> > >
> > > - **Q1: Fine-grained results of MMMU**
> > >
> > > In our previous response, we highlighted differences in fine-grained results for the “Overall-Humanities and Social Science” category. As suggested, we now provide the full set of subcategory results in the table below. These results show clear differences at the fine-grained level, **confirming that the models behave differently across subcategories**. However, due to the averaging across a limited number of samples within a category, it is possible for the overall scores to appear identical. We acknowledge this may cause confusion and will include all fine-grained results in the appendix for clarity in the revised paper.
> > >
> > > |  | **Base MLLM** | **Merged Model** |
> > > | --- | --- | --- |
> > > | Overall-Art and Design | 70.0 | 70.0 |
> > > | Art | 66.7 | 66.7 |
> > > | Art_Theory | 83.3 | 83.3 |
> > > | Design | 86.7 | 86.7 |
> > > | Music | 43.3 | 43.3 |
> > > | Overall-Business | 44.0 | 44.0 |
> > > | Accounting | 33.3 | 33.3 |
> > > | Economics | 53.3 | 53.3 |
> > > | Finance | 43.3 | 43.3 |
> > > | Manage | 40.0 | 40.0 |
> > > | Marketing | 50.0 | 50.0 |
> > > | Overall-Science | 42.0 | 42.0 |
> > > | **Biology** | **40.0** | **43.3** |
> > > | Chemistry | 23.3 | 23.3 |
> > > | **Geography** | **43.3** | **46.7** |
> > > | **Math** | **46.7** | **40.0** |
> > > | Physics | 56.7 | 56.7 |
> > > | Overall-Health and Medicine | 56.7 | 56.7 |
> > > | **Basic_Medical_Science** | **50.0** | **63.3** |
> > > | **Clinical_Medicine** | **73.3** | **70.0** |
> > > | **Diagnostics_and_Laboratory_Medicine** | **36.7** | **40.0** |
> > > | Pharmacy | 60.0 | 50.0 |
> > > | **Public_Health** | **63.3** | **60.0** |
> > > | Overall-Humanities and Social Science | 66.7 | 66.7 |
> > > | **History** | **70.0** | **66.7** |
> > > | Literature | 86.7 | 86.7 |
> > > | **Sociology** | **56.7** | **60.0** |
> > > | Psychology | 53.3 | 53.3 |
> > > | Overall-Tech and Engineering | 37.1 | 37.1 |
> > > | Agriculture | 46.7 | 46.7 |
> > > | Architecture_and_Engineering | 20.0 | 20.0 |
> > > | Computer_Science | 53.3 | 53.3 |
> > > | Electronics | 30.0 | 30.0 |
> > > | Energy_and_Power | 36.7 | 36.7 |
> > > | Materials | 36.7 | 36.7 |
> > > | Mechanical_Engineering | 36.7 | 36.7 |
> > > | Overall | 50.7 | 50.7 |
> > > - **Q2: Effectiveness of IP merging**
> > >
> > > We find that IP merging gives better improvements when a stronger base MLLM is combined with a more capable math LLM. For example, LLaVA 1.5 7B is an early MLLM with weaker reasoning ability. In our experiments, this model was merged with **Tora**, a math LLM that, while effective, lags behind more recent models such as DeepSeek-distilled models. As a result, the gains were small, such as 0.7 on WeMath and MathVision. In contrast, newer models like Qwen2.5 VL and InternVL3, when merged with stronger math models, show larger improvements. For example, Qwen2.5 VL improves by 2.4 on WeMath after merging. We will add discussions of this in the revised paper.
> > >
> > > - **Q3: Reproducibility**
> > >
> > > The main reasons for the performance difference are due to the input pixel constraint and hardware differences. To prevent out-of-memory issues when running the Qwen models for MathVista on 3090, we used the official recommended settings of min_pixels = 256×28×28 and max_pixels = 1280×28×28 to optimize memory usage [1]. However, it may reduce the input resolution and lead to performance differences. When running the same setting on A800, we found the results will be just marginally different, 55.4 vs 54.7, due to hardware differences (different architectures, different precision). **In fact, if we remove the pixel constraint and run the evaluation on GPUs like A800 with larger memory, we obtain 58.4 for the base model and 61.8 for the IP merged model**. As we run most experiments on 3090 (LLava-based models, etc), we applied the same configuration and environment to all merged models to ensure consistency and fair evaluation.
> > >
> > > We fully acknowledge the reviewers’ concern regarding the relatively lower reproduced performance. We emphasize that the lower reproduced scores were not intentional. In fact, on some datasets,  our reproduced scores are even higher than the officially reported values. For example, on the MathVision dataset, our reproduced score was **17.2**, compared to the official **16.3**.  For fair comparison, we use the reproduced results in the paper. To enhance transparency and address your concerns, we will clearly mark reproduced and official scores side-by-side in the revised version and add more implementation details in the appendix to clarify any discrepancies. Thank you again for your valuable feedback.
> > >
> > > [1] Wang et.al, Qwen2-VL: Enhancing Vision-Language Model's Perception of the World at Any Resolution, 2024

---

> > > > ### Comment · Reviewer_XRHa · 2025-08-07
> > > >
> > > > Thank you for your clarification.
> > > >
> > > > The response to Q2 presents a meaningful finding. I believe that the initial version of this work may have placed too much emphasis on methodological arguments while lacking sufficient experimental exploration. However, through discussion, I now recognize the practical value of the method. **Please include all of the experimental results in the final version.**
> > > >
> > > > I appreciate the authors' efforts during this process.
> > > >
> > > > I will raise my score to 4.

---

> > > > > ### Author Response · Authors · 2025-08-07
> > > > >
> > > > > We sincerely thank the reviewer for the thoughtful feedback. We appreciate your constructive suggestions and will ensure the final version reflects these improvements as kindly suggested by you.

---

> > > > > ### Author Response · Authors · 2025-08-09
> > > > > **Final rating reminder**
> > > > >
> > > > > We sincerely thank you once again for your constructive feedback and for agreeing to raise the rating to a positive score. As the review process nears completion, we would greatly appreciate it if you could kindly submit your final rating at your earliest convenience.

---

### Official Review · Reviewer_Vh1P · 2025-07-02

**Clarity:** 3
**Significance:** 3
**Originality:** 3
**Rating:** 4
**Confidence:** 5

**Summary:**

This article attempts to address the issue of merging parameters between Large Language Models (LLMs) and Multimodal Large Language Models (MLLMs) to integrate the reasoning capabilities of an LLM focused on mathematical reasoning into a general MLLM, thereby enhancing the MLLM's ability to handle visual math-related problems. The authors identify two key challenges: how to select appropriate parameters and how to better align them. In response, the authors propose the IP-Merging method, which requires no additional tuning. Moreover, this method not only achieves promising results but also maintains the original capabilities of the MLLM across other tasks.

**Questions:**

Please see weaknesses.

**Ethical Concerns:**

["NO or VERY MINOR ethics concerns only"]

**Final Justification:**

The rebuttal addressed my main concerns with additional experiments on code LLMs, trained visual encoders, and the MathGlance benchmark, all showing consistent gains. The figure issue appears to be a rendering problem. Given the method’s simplicity, training-free nature, and positive results without harming general capabilities, I maintain my borderline accept rating.

**Quality:**

3

**Strengths And Weaknesses:**

Strengths: The writing is fluent, and the analysis of the adjustments to the problem is insightful. The proposed method achieves excellent results without affecting the inherent capabilities of the MLLM.

Weaknesses:

1.Can this method be extended to models other than Math LLMs, such as code LLMs?

2.There are many methods that have found training the visual encoder together can yield good results (e.g., Cambrian-1). Has the author considered this scenario where the visual encoder has been trained?

3.You can consider the results of the benchmark mentioned in MathGlance (https://arxiv.org/pdf/2503.20745) to provide a more insightful assessment.

4.There is some misalignment of text in Figure 2.

---

> ### Author Rebuttal · Authors · 2025-07-31
>
> Thanks a lot for your insightful comments. Here, we address your concerns and questions point by point.
>
> - **W1: Extend to other models**
>
> **Yes, the method can be extended to other models like code LLM.** In our paper, we tried merging CodeLlama and Tora-code with the LLaVA model, which improves MathVista performance from **25.2% to 26.3% and 28.2%, respectively**. It shows that code LLMs can contribute to multi-modal reasoning tasks. The experimental results are shown in  **Table 10 of Appendix.**
>
> - **W2: Trained visual encoder**
>
> We are thankful for your insights. In our work, we focus on developing a train-free method to enhance the reasoning performance of VLMs, therefore we do not retrain the visual encoder in the paper. Nevertheless,  IP-merging also works  effectively when the base MLLM has a trained vision encoder.** We test the Cambrian-8B model and merge the Deepseek-R1-distilled model (LLama-3-based reasoning LLM). We illustrate the results on MathVista, showing that IP merging can further enhance the reasoning performance.
>
> | Model | **MathVista** |
> | --- | --- |
> | Cambrian-8B | 46.8 |
> | **+IP megring** | **48.8** |
> - **W3: MathGlance benchmark**
>
> Thank you for your great suggestion. We conducted experiments on the MathGlance Benchmark using Qwen2.5-VL-7B-Instruct  as the base MLLM and the Deepseek-R1-distilled-Qwen-7B as reasoning LLM. After merging the LLM,  the geometry reasoning performance of the base MLLM is further enhanced.
>
> | Model | **Plane Geometry** | **Solid Geometry** | **Graphs** |
> | --- | --- | --- | --- |
> | Qwen2.5-VL-7B-Instruct | 44.5 | 67.8 | 64.7 |
> | **+IP Merging** | **45.8** | **69.0** | **67.4** |
> - **W4: Misalignment of the text**
>
> Thanks. However, we cannot find the misalignment in Figure 2. The figure appears correctly formatted on our end (We previously found that if an old version of a browser or a PDF reader is used, it may not show the figure correctly).

---

> > ### Comment · Reviewer_Vh1P · 2025-08-05
> > **Official Comment by Reviewer Vh1P**
> >
> > Thank you for the proactive rebuttal, which addressed my concerns. Considering other reviewers’ feedback and the contribution to the community, I will keep my positive score.

---

> > > ### Author Response · Authors · 2025-08-07
> > >
> > > Thank you for your thoughtful response. We are glad that we addressed your concerns, and we sincerely appreciate your positive comments.

---

### Official Review · Reviewer_F8Nd · 2025-07-03

**Clarity:** 3
**Significance:** 3
**Originality:** 2
**Rating:** 4
**Confidence:** 4

**Summary:**

This paper addresses the challenge of enhancing the mathematical reasoning abilities of multi-modal large language models (MLLMs) by leveraging the expertise of specialized, text-only math LLMs. The authors explore whether this transfer of knowledge can occur without any expensive fine-tuning, proposing that a significant gap in the parameter spaces of these models is the primary obstacle to direct merging.

To overcome this, the paper introduces a tuning-free method named IP-Merging. This approach first identifies the most relevant parameters for the task through subspace similarity analysis and then projects them from the Math LLM into the MLLM's subspace to ensure better alignment. The authors validate their method on several visual math reasoning benchmarks, showing that it improves performance on the target task while successfully preserving the model's general capabilities.

**Questions:**

1. This work focuses exclusively on mathematical reasoning. Have you considered whether IP-Merging is equally effective for transferring other specialized, text-based abilities, such as code generation, legal knowledge, or medical question-answering? Does the success of the method depend on any particular properties of the task being transferred?


2. The similarity threshold `Sω` is a key hyperparameter. Instead of a fixed threshold, have you considered a more adaptive selection strategy? For instance, one could select a certain top percentage of layers based on their similarity score distribution, which might make the method more robust across different pairs of models.

3. The performance on the GPS subset is notably poor compared to Ties Merging, and your explanation for this is quite general. Could you provide a more detailed error analysis for this task? For example, does the model fail more on visual perception of shapes or on applying geometric reasoning? This would help clarify the root cause of this limitation.

4. To better demonstrate the method's utility, it would be valuable to see experiments on more recent, powerful MLLMs like Qwen2.5-VL and InternVL3. Since these models already have strong baseline reasoning abilities, showing that IP-Merging can still provide a meaningful boost would be a critical test of its practical value.

5. The claim that IP-Merging preserves general abilities is supported only by the MMMU benchmark. To make this claim more robust, could you provide evaluation results on other general multi-modal benchmarks, such as MMBench or TextVQA?

**Ethical Concerns:**

["NO or VERY MINOR ethics concerns only"]

**Final Justification:**

I thank the authors for their detailed rebuttal and additional experiments. The paper's core idea of tuning-free knowledge transfer is well-motivated and presents a clear contribution. However, my primary concerns persist. The performance degradation on geometric reasoning tasks (the GPS subset) and the modest gains on more capable MLLMs are unresolved issues that temper the method's overall impact. For these reasons, I will maintain my original score.

**Limitations:**

yes

**Paper Formatting Concerns:**

I did not notice any major formatting issues. The paper appears to adhere to the NeurIPS 2025 formatting instructions regarding page limits, style, and structure.

**Quality:**

2

**Strengths And Weaknesses:**

The work's main strength is its focus on a highly relevant and practical problem: finding low-cost ways to enhance powerful models with specialized skills. The diagnostic analysis presented is insightful, providing a clear and data-driven motivation for why simple model merging fails and why a more sophisticated approach is necessary. The experimental validation is also quite solid, especially the inclusion of a general-purpose benchmark to confirm that the method does not cause catastrophic forgetting, which is a critical aspect for any model merging technique.

However, the paper has some notable weaknesses that temper its overall impact. The most significant limitation is that the proposed method is currently restricted to models that share an identical architecture and size. This severely narrows its practical applicability, as many real-world use cases would involve transferring knowledge between models of different scales or families. Furthermore, the methodological contribution feels somewhat incremental, as it primarily combines well-established techniques like SVD and task vector arithmetic in a novel application, rather than introducing fundamentally new concepts.

Beyond these broader concerns, some specific experimental results raise further questions. According to Table 2, the merging of LLaVA-1.5-7B shows a very large performance gap (18.8 points) on the GPS subset of MathVista when compared to Ties Merging. This result casts doubt on the effectiveness of IP-Merging for geometric problems. The authors' explanation that this is because GPS requires multiple steps of geometry reasoning is insufficient; if the MLLM truly absorbed the Math LLM's reasoning ability, it should show improvement in this domain. Additionally, the method has not been validated on stronger MLLMs like Qwen-2.5-VL or InternVL 3. For example, InternVL 3 already achieves high performance of 39.8 on MathVerse and 71.6 on MathVista. Verifying that IP-Merging still provides a meaningful benefit to these highly capable models would be very helpful for testing the method's validity.

---

> ### Author Rebuttal · Authors · 2025-07-31
>
> We appreciate your insightful comments. Here, we address your concerns and questions point by point.
>
> - **Q1: Transfer other text-based abilities**
>
> Thank you for the inspiring question. Math reasoning has been one crucial ability of LLMs and MLLM, hence we focus on math reasoning in our paper.  Essentially, our method relies on using a specialized model and its task vector to transfer capabilities from LLMs to corresponding MLLMs. This setup is not restricted to math reasoning and could be applied to other specialized domains such as code generation. However, we have not found an appropriate testing benchmark to evaluate the effect of the code generation ability of MLLMs. In our paper, we also made attempts to merge the code model to enhance math reasoning in Table 10 in the appendix, demonstrating its effectiveness. For the medical image question answering, we believe the proposed IP-merging might be useful, where both task-specific knowledge and visual understanding are required. In case that more benchmarks and corresponding specialized models are available, we are happy to investigate extending our method in these domains.
>
> - **Q2: Adaptive&fixed similarity threshold**
>
> While we fully understand your concern, we suggest setting the threshold directly (we find 0.4 performs well in our experiments).   Our justifications are as follows. Our goal is to identify crucial parameters between models using similarity scores.  We directly use a similarity threshold based on the two considerations: (1) **A fixed similarity threshold ensures that only highly related parameters are merged.** **In contrast, a top-k% approach selects a fixed number of parameters regardless of their similarity. Some layers might include low-similarity parameters, which may not benefit from merging.**  For some model pairs, the top-k% **parameters may have high similarity, while for others, they might be relatively dissimilar. This makes the threshold harder to tune across different model pairs. (2) **Similarity scores are normalized,** so the fixed threshold is naturally robust across different model pairs.  For your reference, we also conduct experiments of selecting the top-k percentage of parameters based on the score distribution. As shown in the following table, we find that the performance is not as robust compared to directly setting the threshold (0.4).  Although the two approaches are similar, directly setting the threshold is a more effective way for our method.
>
> | Adaptive threshold | 5% | 10% | 15% | 20% | 25% | 30% |
> | --- | --- | --- | --- | --- | --- | --- |
> | Corrspondingsimilarity threshold | 0.45 | 0.25 | 0.16 | 0.12 | 0.10 | 0.08 |
> | IP merging  | 26.1 | 26.1 | 25.0 | 26.7 | 25.9 | 26.8 |
> - **Q3: Poor performance on GPS**
>
> Thank you for your comments.  Following your suggestions, we examine failure cases in MathVista,  and find that the IP-merged LLaVA model may struggle to recognize key visual information in the image, leading to wrong predictions. The cases below suggest that the IP-Merged model’s performance is hindered by weakened visual understanding, particularly in tasks requiring recognition of geometric information. In the revision, we will add one section of case studies for more discussions.
>
> | Cases | Questions | Comments |
> | --- | --- | --- |
> | MathVista-id:143,
> image: images/143.jpg | In the parallelogram ABCD, line CE bisects ∠BCD and intersects edge AD at point E. Given DE = 3.0, what is the length of AB?  | **This question requires the model to recognize parallelogram image and use properties of angle bisectors and parallelograms to solve problem.** |
> | Ties-Merging | Prediction: 3 **(True)** |  |
> | IP-Merging | Prediction: 1 **(False)** | Failure on visual perception (The model fails to recognize which is edge AB） |
> | MathVista-id:59,
> image:images/59.jpg | In the figure, AB is the diameter of circle ⊙O, and CD is a chord. Given that ∠ADC = 26°, what is the degree of ∠CAB?  | **This question requires the model to locate the right triangle in the circle image and make predictions use the relationships between diameters, chords, and inscribed angles.** |
> | Ties-Merging | Prediction: The correct answer is 64 **(True)** |  |
> | IP-Merging | Prediction : The correct answer is 26 **(False)** | Failure on visual perception (The model does not locate the triangle in the circle accurately） |
> - **Q4-Q5: Experiments of recent MLLMs and other general multi-modal benchmarks**
>
> We are thankful for your suggestions. Following your kind suggestion, we further conduct experiments using recent MLLMs Qwen2.5-VL-7B-Instruct and InternVL3-8B-Instruct on more benchmarks including MathVision,  Dynamath, Wemath and one additional general VQA benchmark **TextVQA** (Due to the limited rebuttal window, we do not run MMBench). We use Deepseek-R1-distilled-Qwen-7B as the math LLM.  As shown in the table, our methods do enhance the reasoning performance of stronger baseline models while maintaining the general reasoning abilities.
>
> | Model | **Mathverse** | **MathVista** | **MathVision** | **Dynamath** | **Wemath** | **TextVQA** |
> | --- | --- | --- | --- | --- | --- | --- |
> | Qwen2-VL-7B-Instruct | 24.8 | 55.4 | 17.2 | 40.8 | 23.6 | 83.8 |
> | +IP Merging | **28.5** | **60.2** | **19.1** | **41.0** | **24.4** | **84.0** |
> | LLava-1.5-7B | 11.3 | 25.2 | 11.1 | 15.4 | 6.5 | 47.50 |
> | +IP Merging | **15.4** | **28.4** | **11.8** | **16.1** | **7.2** | **47.52** |
> | Qwen2.5 VL-7B-Instruct | 46.6 | 68.0 | 24.7 | 51.9 | 31.1 | 84.8 |
> | +IP merging | **46.8** | **69.7** | **26.3** | **52.5** | **33.5** | **84.9** |
> | InterVL3-8B-Instruct | 38.8 | 66.1 | 24.9 | 50.7 | 31.6 | 81.9 |
> | +IP meging | **39.0** | **67.6** | **25.2** | **51.4** | **33.0** | **82.3** |

---

### Note · Authors · 2025-08-13

We sincerely thank all reviewers for their constructive feedback and thoughtful questions. In our rebuttal, we have addressed each concern in detail, clarified experimental settings, added results on stronger and more diverse MLLM baselines, expanded benchmark coverage, and provided additional error analyses and case studies.

**All reviewers are now positive about the paper**, and we especially appreciate reviewer **XRHa** raising their score from 2 to 4, and reviewers **F8Nd** and **Vh1P** have indicated they are inclined to maintain their scores of 4. We believe these clarifications, new experiments, and planned revisions will further strengthen the paper.

---

### Decision · Program_Chairs · 2025-09-17

**Decision:**

Accept (poster)

**Comment:**

## Summary

This paper studies whether multi-modal large language models (MLLMs) can gain mathematical reasoning ability from specialized math LLMs without additional training. The authors note that direct merging of parameters between the two types of models is limited by differences in their parameter spaces. To address this, they introduce a method called IP-Merging. This approach first identifies the parameters most related to reasoning in both models, then projects them into the MLLM subspace to reduce misalignment, and finally merges them. The method is tuning-free since it adjusts parameters directly rather than relying on gradient-based training. Experiments show that IP-Merging improves math reasoning in MLLMs while preserving their general performance.

## Decision

This paper addresses an important problem: finding low-cost ways to enhance powerful models with specialized skills. The experiments are thorough and well executed, and the results are convincing. The writing is clear, and the paper is easy to follow. In addition, the paper provides a detailed empirical analysis explaining why conventional merging methods are ineffective, clearly demonstrating the existence of a “parameter space gap.”

The proposed IP-Merging method is interesting, though somewhat incremental. During the review process, several concerns and criticisms were raised, but the authors effectively addressed them in the rebuttal. I recommend that the authors incorporate the additional results presented in the rebuttal to further strengthen the paper.

Overall, the reviewers were positive about this work. I recommend the paper for acceptance, as it should be of interest to the ICLR community.